# NGC1818 unveils the origin of the extended main-sequence turn-off in young Magellanic Clouds clusters

Giacomo Cordoni [1] ✉, Antonino P. Milone[1,2], Anna F. Marino [3], Michele Cignoni[4,5], Edoardo P. Lagioia[1], Marco Tailo[6], Marília Carlos [1], Emanuele Dondoglio[1], Sohee Jang[1], Anjana Mohandasan [1] & Maria V. Legnardi [1]

The origin of young star clusters represents a major challenge for modern stellar astrophysics. While stellar rotation partially explains the colour spread observed along main-sequence turn-offs, i.e. where stars leave the main-sequence after the exhaustion of hydrogen in their core, and the multiple main sequences in the colour-magnitude diagrams of stellar systems younger than approximately 2 Gyr, it appears that an age difference may still be required to fulfill the observational constraints. Here we introduce an alternative approach that exploits the main-sequence turn-on, i.e. the point alongside the colour-magnitude diagram where pre-main-sequence stars join the main-sequence, to disentangle between the effects of stellar rotation and age to assess the presence, or lack thereof, of prolonged star formation in the approximately 40-Myr-old cluster NGC1818. Our results provide evidence for a fast star formation, confined within 8 Myr, thus excluding age differences as responsible for the extended main-sequence turn-offs, and leading the way to alternative observational perspectives in the exploration of stellar populations in young clusters.

The paradigm that young star clusters are the simplest stellar structures in the Universe has been challenged by the discovery of multiple sequences of stars alongside specific features in the colour–magnitude diagrams (CMDs). Specifically, until a decade ago, young clusters were considered prototypes of simple stellar populations, where all stars share the same age. This assumption was supported by state-of-the-art CMDs where the photometric sequences of young clusters were similar to single isochrones (e.g., ref. [1]).

When the Hubble Space Telescope (HST) came into the picture, it revealed unexpected features in young Magellanic Cloud star clusters like extended main-sequence turn-offs (eMSTOs[2]), and split main-sequences (MSs[3]). Rather than being exceptional features, extensive observations revealed the nearly ubiquitous nature of the eMSTOs in young clusters[4–6], which are detected both in massive and in low-mass clusters[7,8].

The eMSTOs have been immediately interpreted as the signature of prolonged star formation (e.g., refs. [5,9]), as the age spread inferred from it can range from a few tens of Myr to ~500 Myr. Therefore, young Magellanic Clouds clusters were immediately considered the missing link to understanding the evolution of old globular clusters (GCs) with multiple stellar populations, i.e. star-to-star light-element abundance variations[10,11].

However, the hypothesis that young star clusters host multiple stellar generations has been challenged by the idea that a spread in

[1]Dipartimento di Fisica e Astronomia "Galileo Galilei", Università degli Studi di Padova, Padova IT-35122, Italy. [2]Istituto Nazionale di Astrofisica, Osservatorio Astronomico di Padova, Padova IT-35122, Italy. [3]Istituto Nazionale di Astrofisica, Osservatorio Astrofisico di Arcetri, Firenze IT-50125, Italy. [4]Dipartimento di Fisica "E. Fermi", Università degli Studi di Pisa, 56127 Pisa, Italy. [5]Istituto Nazionale di Fisica Nucleare, 56127 Pisa, Italy. [6]Dipartimento di Fisica e Astronomia Augusto Righi, Università degli Studi di Bologna, I-40129 Bologna, Italy. ✉e-mail: giacomo.cordoni@unipd.it

stellar rotation rate can mimic the effects of an age difference in a cluster formed of coeval stars[12–14] and by the lack of prolonged star formation in young star clusters of starburst galaxies[15–18]. Evidence that stars with different rotation rates are associated with eMSTOs and split MSs is provided by direct spectroscopic measurements[19–21], by the presence of fast-rotating Be stars in the eMSTO[8,22], and by the correlation between the age inferred by the eMSTO width and the cluster age[23].

Although it is now widely accepted that rotation plays an important role in shaping the CMD of young clusters, it is not clear whether the eMSTO is entirely due to stellar rotation or if it is the result of the combined effect of age and rotational velocity in stars[24]. In the present work, following the results of[3], we a priori exclude metallicity and/or light-element abundance variations as possible responsible for eMSTo and split-MSs. Moreover, while stellar rotation diminishes but does not eliminate age spreads (refs. 24, 25 and references therein), a mix of age variation and rotation provides a good match with the studied CMDs (refs. 26, 27 and references therein). Hence, the debate is still entertained on whether the young clusters host multiple stellar generations or if the eMSTO and split-MSs, or if other physical mechanisms play a role.

In this work, we adopt an alternative approach to assess whether young clusters host stars of different ages or not, by exploiting the turn-on feature where the pre-main-sequence reaches the main sequence[28]. The turn-on, involving a different range of masses with respect to the turn-off, represents an independent clock[29–31] and has never been used to determine the star formation history of young Magellanic Clouds clusters with eMSTO and split MSs. Our investigation of the turn-on luminosity function of NGC1818, a ~40 Myr-old low-mass star cluster in the Large Magellanic Cloud which exhibits a split-MS and an eMSTO, supports a fast star formation that lasted at most 8 Myr. The small age variation inferred excludes, once and for all, that age spread is the main responsible for the extended turn-off.

## Results

### NGC 1818 the Rosetta Stone to disentangle age and rotation

To address this astrophysical issue, the Hubble Space Telescope has collected deep images of the young LMC cluster NGC1818 (details of the observations are presented in Table 1), where the eMSTO has been well studied by previous work[21,22]. NGC1818 represents the perfect target to unveil the origin of eMSTOs in young Magellanic Clouds clusters, as it is very young and shows the typical features of young Magellanic Clouds clusters' CMDs. Although NGC1818 is not one of the most-massive clusters with multiple sequences (mass, $M = 3 \times 10^4 M_\odot$, ref. 22), our choice is supported by the evidence that the main properties of young clusters, like the presence of the eMSTO, the relative numbers of blue- and red-MS stars, and the colour width of the multiple sequences, do not depend on cluster mass[22]. Hence, it is reasonable that the conclusion on the origin of the young star cluster NGC1818 can be extended to the other clusters with eMSTO and split MS.

The resulting three-colour image of stars around the cluster centre is presented Fig. 1a and is obtained from the same dataset (Supplementary Data 1) used to derive the CMD in the F606W and F814W filters, mounted on the Wide Field Camera 3 (WFC3) on board

of HST, and the luminosity function (LF) plotted in Fig. 1b, d, respectively. Figure 1c shows the CMD of stars in the reference field, used to quantify and correct for field stars contamination. The exquisite photometry obtained from HST images allowed to approach the pre-main-sequence (pre-MS) of NGC1818, whose brightest point, i.e. the MS turn-on, is highlighted by the peak of the LF around a magnitude in F814W = 23.5, indicated by the black solid line in Fig. 1b–d.

### An apparent large age spread in NGC1818 inferred from the main-sequence turn-off

The presence of multiple sequences in the CMD of NGC1818 becomes evident when we move from the optical regime, where it resembles a single isochrone, to CMDs made with ultraviolet (UV) photometry. Specifically, the HST WFC3/UVIS F336W and F814W filters reveal a double MS and an extended main-sequence turn-off as highlighted in Fig. 2.

Following the idea that such features are the result of prolonged and/or multiple star formation episodes alone, we exploited grids of non-rotating isochrones of different ages to estimate the age distribution of turn-off stars and found that NGC1818 stars span a wide age interval from ~40 Myr up to 100 Myr (see 'Methods', subsection Age determination from the main-sequence turn-off for details). Hence, the age distribution shown in Fig. 2c has been derived by assuming that the colour and magnitude broadening of MSTO stars is entirely due to age variations and that stellar rotation does not affect the eMSTO phenomenon.

The age distribution of NGC1818 stars makes it tempting to relate the eMSTO phenomenon to the multiple populations in old GCs. Moreover, the predominance of young stars in NGC1818 (ages of roughly 50 Myr) would recall what is observed in most GCs, where the majority of stars belong to the second population[32].

However, the age distribution inferred in Fig. 2 is derived from isochrones that do not account for stellar rotation, represented in Fig. 2b, and the evidence that coeval stars with different rotation rates can mimic an age spread[12] makes it virtually impossible to discriminate the contribution of age and rotation in shaping the eMSTO. As a consequence, no firm conclusion on the origin of eMSTOs and split-MSs can be inferred from the eMSTO alone.

### The true star-formation history of NGC1818 unveiled by the main-sequence turn-on

Here we follow an alternative approach to disentangle between age and rotation effects and constrain the alleged age spread of the stellar populations in a young cluster. The adopted method, which is based on the turn-on, has allowed to detect age variations down to the Myr scale in several Magellanic Cloud star-forming regions and young open clusters[29,31,33,34], but has never been used, so far, to investigate the star formation history of young Magellanic Clouds clusters with eMSTO and split MSs.

The turn-on is the point along the CMD where pre-MS stars join the main-sequence. It is populated by the most-massive pre-MS stars, for which the time spent in the pre-MS phase corresponds to the stellar age. Hence, the turn-on luminosity in a star cluster depends on the age of its stellar populations (see e.g., refs. 28, 35–37), as shown in Fig. 3a. At odds with the MS turn-off, whose luminosity is strongly affected by

**Table 1 | Description of the images collected through the HST Ultraviolet and Visual channel of the Wide-Field Camera 3 used in this work**

| Filter | Date | N EXPTime | Programme | Dataset |
|--------|------|-----------|-----------|---------|
| F336W | October 29 2015 | 10 s + 100 s + 790 s + 3 × 947s | 13727 | https://archive.stsci.edu/proposal_search.php?mission=hst&id=13727 |
| F606W | June 29 2020 | 724 s + 2 × 772s | 15945 | https://archive.stsci.edu/proposal_search.php?mission=hst&id=15945 |
| F814W | February 2 2017 | 90 s+666 s | 14710 | https://archive.stsci.edu/proposal_search.php?mission=hst&id=14710 |
| F814W | June 29 2020 | 81 s + 2 × 795s | 15945 | https://archive.stsci.edu/proposal_search.php?mission=hst&id=15945 |

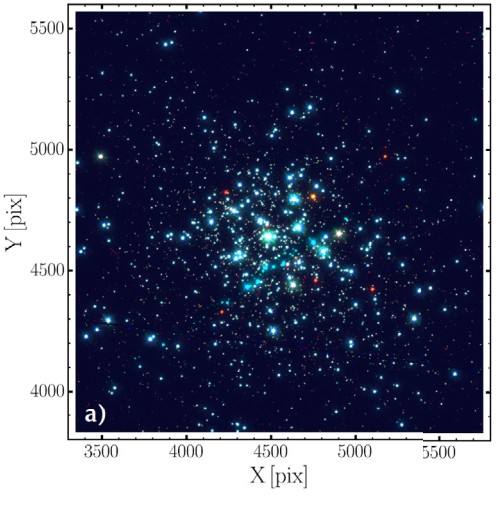

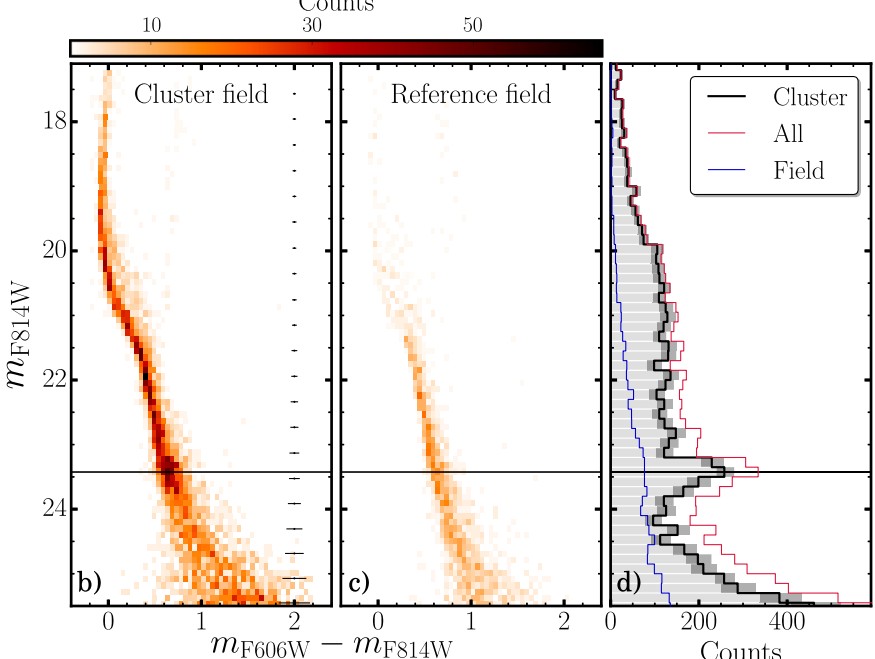

**Fig. 1 | Hubble Space Telescope view of NGC 1818 and its colour–magnitude diagram. a** Three-colour image of a portion of the analysed Hubble Space Telescope field around the centre of NGC1818. The pixel scale is 0.04 arcsec/pix. **b, c** Colour–magnitude diagram of cluster-field and reference-field stars, respectively. The top colourbar indicates stellar counts. Photometric uncertainties, inferred from artificial star tests as discussed in the 'Methods' section, subsection artificial stars, are indicated on the right of (**a**). **d** The field subtracted Luminosity Function of NGC1818 is represented with the solid black line, while the blue line indicates the contribution of reference-field stars and the red line the unsubtracted cluster-field Luminosity Function. The horizontal lines mark the magnitude of the MS Turn on in all three panels.

stellar rotation[13,14], the turn-on of stellar populations with different rotation rates share nearly identical luminosity as inferred from MIST isochrones[38–40]. The fact that the turn-on luminosity is strongly dependent on cluster age while poorly affected by rotation, makes it an exquisite clock to precisely date the stellar populations in these clusters. In the following, we compare the age distributions inferred from turn-on and turn-off stars to derive a star-formation history of NGC1818 by using a method that is not affected by stellar rotation. Results will allow us to understand whether multiple or prolonged star formation episodes are responsible for the eMSTO.

The turn-on feature is visible as a stellar overdensity in the CMD and is easily detectable as a narrow peak in the Luminosity Function, as illustrated in Fig. 3b–e. We illustrate in the 'Methods', subsection Age

determination from the main-sequence turn-on, how the LF changes with cluster age, and show that the luminosity of the turn-on peak decreases for older ages. Clearly, multiple peaks in the LF of a star cluster would be the signature of multiple bursts of star formation, whereas a simple stellar population corresponds to a narrow single peak. Similarly, prolonged star formation would produce a broad peak in the LF.

A visual inspection of the LF of NGC1818, indicated by the black line in Fig. 1d, reveals one prominent peak at F814W magnitude of roughly 23.5 (black horizontal lines in Fig. 1b–d), which will be used as a tracer to infer the age distribution of cluster members. By adapting to NGC1818, the method by Cignoni et al.[29–31], we constructed grids of synthetic CMDs composed of different stellar populations with

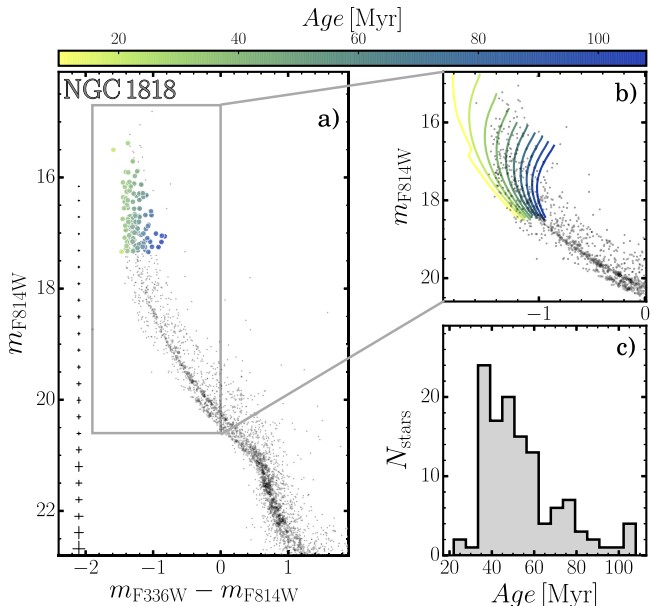

**Fig. 2 | Extended main-sequence turn-off in NGC1818. a** Colour–magnitude diagram of NGC1818 from optical and ultraviolet magnitudes, i.e. WFC3/UVIS F814W and F336W. turn-off stars are colour-coded according to their inferred ages, as indicated in the top colourbar. Photometric uncertainties, shown on the left, are computed from artificial star tests, as discussed in the 'Methods' section, subsection artificial stars. **b** Zoom of the turn-off region. The solid lines superimposed on the CMD are non-rotating isochrones, computed with the Padova models, with ages ranging from 10 to 100 Myr in steps of 10 Myr. **c** Age distribution of turn-off stars derived from the isochrone interpolation.

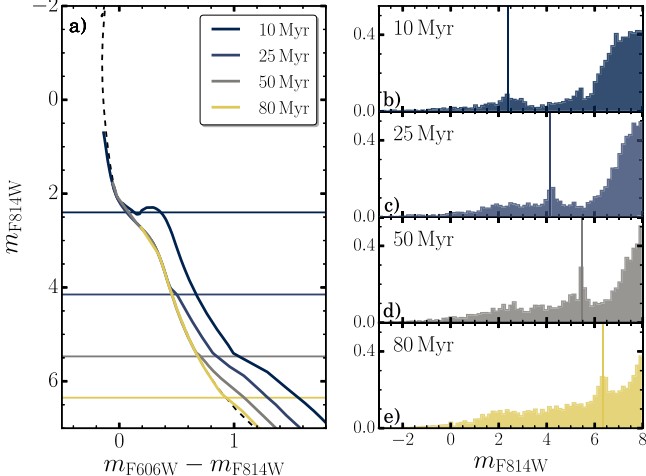

**Fig. 3 | The turn-on to date stellar populations. a** Isochrones different ages, from 10 to 80 Myr, computed with the Padova models[47]. The age of each isochrone is indicated in the top right legend. The black shaded line indicates the zero-age main-sequence (ZAMS), while the horizontal lines mark the magnitude of the turn-on of each stellar population. **b–e** Luminosity function of stellar populations with the same age as the isochrones shown in (**a**). The vertical lines mark the location of the main-sequence turn-on distinguishable as a peak in each LFs. The age of each stellar populations is indicated in the top left corner.

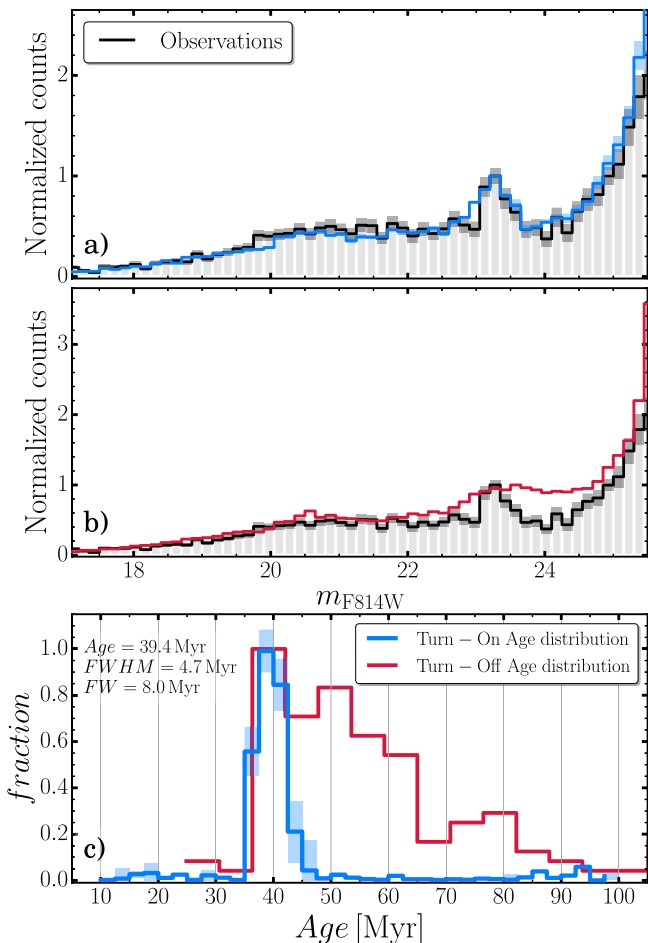

**Fig. 4 | The Luminosity Function of NGC1818 and its age distribution.**
**a** Observed Luminosity Function of main-sequence stars of NGC1818 is shown in solid black, while the uncertainties are indicated by the dark-grey shaded regions. The blue line corresponds to the best-fit LF, with uncertainties indicated by the azure-shaded regions. **b** As in (**a**), the black line indicates the observed LF, whereas the red line represents the LF inferred from the age distribution inferred from the eMSTO (panel c of Fig. 2). The y-axis of (**a**) and (**b**) have been normalized so that the height of the turn-on peak corresponds to 1. **c** Age distributions of NGC1818 inferred from the turn-off (red) and the turn-on (blue). The shaded regions represent the inferred uncertainties. The top-left inset shows the mean, FWHM and full width, in Myr, of the least-square best fit Gaussian function.

uncertainties, inferred with bootstrap procedure, for observations and simulations, respectively, determined as the 68th percentile of each bin distribution. The red line in Fig. 4b represents the LF derived from the age distribution of turn-off stars, indicated in Fig. 4c with the same colour. The top-left inset in Fig. 4c shows the result of the best-fit Gaussian. The analysis indicates a fast star formation for NGC1818 with a duration of <8 Myr, which corresponds to four times the dispersion of the best-fit Gaussian function, σ. On the other hand, considering the full width at half maximum (FWHM), more commonly used in the literature, would further reduce starburst duration, to 4.7 Myr. In contrast, the age distribution inferred from the eMSTO (shown with a red solid line in Fig. 4c) would provide a smooth LF, which poorly matches the observed ones (Fig. 4b).

## Discussion
Fourteen years after the early discoveries of the eMSTO[2,4], the physical mechanisms that are responsible for this phenomenon are still a matter of heated debate. It is now widely accepted that stellar rotation contributes to the colour broadening of MS stars, but age variations

different ages and compared the resulting LFs with the observed one employing a chi-squared statistical analysis (see 'Methods', subsection Age determination from the main-sequence turn-on for details). The simulated LF that best fits the observations is shown in blue in Fig. 4a and the histogram distribution of stellar ages is represented in Fig. 4c with the same colour. Dark-grey and light-blue shaded regions mark the

are, still, mandatory to quantitatively reproduce the observed eMSTOs[24]. However, as of today, age determinations are based on specific CMD features, such as the eMSTO, which are strongly affected by stellar rotation. Hence, the presence of residual age spread is still controversial[24,26,27].

Results presented in this work, which are based on the methodology developed by Cignoni et al.[29–31] and widely used to infer star formation history of young star clusters, rule out, once and for all, age variations larger than 8 Myr in the eMSTO cluster NGC1818.

Figure 4c reveals a striking discrepancy between the age distributions from the turn-on and the turn-off, shown respectively with blue and red lines, which demonstrate that the apparent wide age spread inferred from the eMSTO is an artefact due to stellar rotation. Clearly, NGC1818 exhibits a single major peak around 40 Myr, which indicates that this cluster has undergone only one star formation episode that lasted no more than 8 Myr.

Hence, the finding of negligible age spreads in young clusters with the eMSTO excludes the possibility that stellar populations with different ages are responsible for the eMSTO.

The James Webb Space Telescope will allow deep near-infrared observations of low-mass stars in the turn-on region, in virtually all young LMC and SMC star clusters with the eMSTO, including populous star clusters like NGC1866 and NGC1850. The resulting deep and accurate photometry, together with a large number of stars would allow to significantly reduce the uncertainties in age-spread determination further constraining the origin of the eMSTO in Magellanic Clouds clusters.

## Methods

### Data and data reduction

To investigate stellar populations in NGC1818 we exploit images collected through the F336W, F606W and F814W filters of the Ultraviolet and Visual channel of the Wide-Field Camera 3 (UVIS/WFC3) on board HST, collected as part of the observing programme GO 15495, P.I Cordoni. The main information on the dataset is provided in Table 1.

Stellar photometry and astrometry of stars in a ~2.7-square arcmin field centred on NGC 1818 are derived from the images corrected for the effects of poor charge transfer efficiency of UVIS/WFC3 (CTE, ref. 41) and the computer programme KS2. KS2 has been developed by Jay Anderson and is the evolution of kitchen_sync, originally written to reduce two-filter ACS/WFC images[42]. It is based on the effective Point Spread Functions (PSF) and follows different recipes to derive the optimal measurement of stars of different luminosities.

KS2 follows an iterative approach to detect and measure stars. First, it calculates the positions and fluxes of bright and isolated stars and subtracts them from the image. This procedure is then repeated for fainter and less isolated stars, allowing to obtain high-quality photometry for all sources, from bright and isolated stars to faint stars in crowded regions. The best photometry and astrometry of bright stars is obtained by measuring the stars in each exposure independently. Measurements inferred from each exposure are then averaged together to get the final photometry and astrometry.

On the other hand, faint stars have low fluxes and their positions and magnitudes cannot be properly inferred in each exposure. Hence, to derive the positions and magnitudes of these faint stars, KS2 combines stellar fluxes from the entire dataset by fixing the average stellar positions from all exposures. Then, after subtracting neighbouring stars, it uses aperture photometry to derive the magnitude of each star. We refer to papers by[43] for further details. We corrected the derived stellar positions for geometric distortions by following the recipe by Bellini and collaborators[44,45] and photometry has been calibrated by adopting the most-updated zero points provided by the STScI webpage.

To investigate stellar populations in NGC1818, we defined a cluster field, which encloses most of the cluster members and comprises stars with radial distance from the centre smaller than 50 arcsec[22].

Such radius corresponds to ~12 parsec, consistent with the radius containing 90% of the total light and well beyond the core radius of 2.5 parsec as determined in ref. 46. We repeated the analysis for different cluster radii, namely 40, 44, 48 and 52 arcsec, finding consistent results. To correct for field stars that contaminate the cluster field, we defined a reference field as a region with radial distance from the cluster centre larger than 75 arcsec. Finally, to account for the different areas of the cluster and reference field and avoid oversubtraction, we scaled the number of stars in the reference field for the ratio of the two areas.

Moreover, verified that reddening variations, if present, are smaller than $E(B-V) = 0.003$ mag, and are much smaller than photometric errors.

### Artificial star tests

To estimate the photometric uncertainties, derive the completeness level of the photometry and generate the simulated CMDs, we performed artificial star tests as in ref. 41. In a nutshell, we generated 100,000 stars with the same spatial distribution as the observed stars, distributed along the fiducial line of the observations in the magnitude range (−5.0, −13.8), in instrumental F814W magnitudes (defined as −2.5 log (flux), where the flux is provided in photoelectrons and is recorded to the reference exposure, which is the one with the longest exposure time).

Specifically, each simulated artificial star has been added to each individual exposure, with its simulated flux and position, and measured it by following the same method used for real stars, i.e. with KS2. We then considered a star as detected if the input and the output position differ by <0.5 pixel and the fluxes by <0.75 mag. Moreover, the analysis has been performed on one star at a time so that artificial stars do not influence each other.

To infer photometric uncertainties, we computed the difference between the output and the input magnitudes and derived the median and the 68th percentile of their distribution in different magnitude bins. This approach accounts for the fact that photometric uncertainties, which are shown on the right of Fig. 1b, mostly depend on stellar luminosity.

Completeness has been calculated by accounting for the dependence of luminosity and crowding on the number of recovered stars. The completeness level has been determined as the fraction of detected stars over input artificial stars within specific magnitude and radial bins.

On average, the completeness ranges from ~100% at a magnitude $m_{F814W} \leq 19$ to ~50% at $m_{F814W} \sim 25.5$. For the luminosity of MS turn-on stars, $m_{F814W} \sim 23.5$, the completeness is roughly 75%.

### The binary fraction of NGC 1818

Unresolved binaries significantly affect the distribution of stars in the CMD of a star cluster and the corresponding LF. As a consequence, an appropriate comparison of simulated and observed star clusters requires accurate determinations of the fraction of binaries.

In a simple stellar population, the position in the CMD of each binary system that is composed of two non-interacting MS stars (MS-MS binaries) depends on the mass of the primary star and on the mass ratio. Here the mass ratio, $q = M_2/M_1$, is the ratio between the mass of the secondary and the primary star, and ranges from 0 to 1, where 0 means that there is no binary system, and 1 that both stars have the same mass.

To investigate the behaviour of MS-MS binaries in NGC1818, we exploit isochrones from the Padova database[47]. The best match with the observed CMD has been obtained by adopting values of $(m-M)_0 = 18.30$

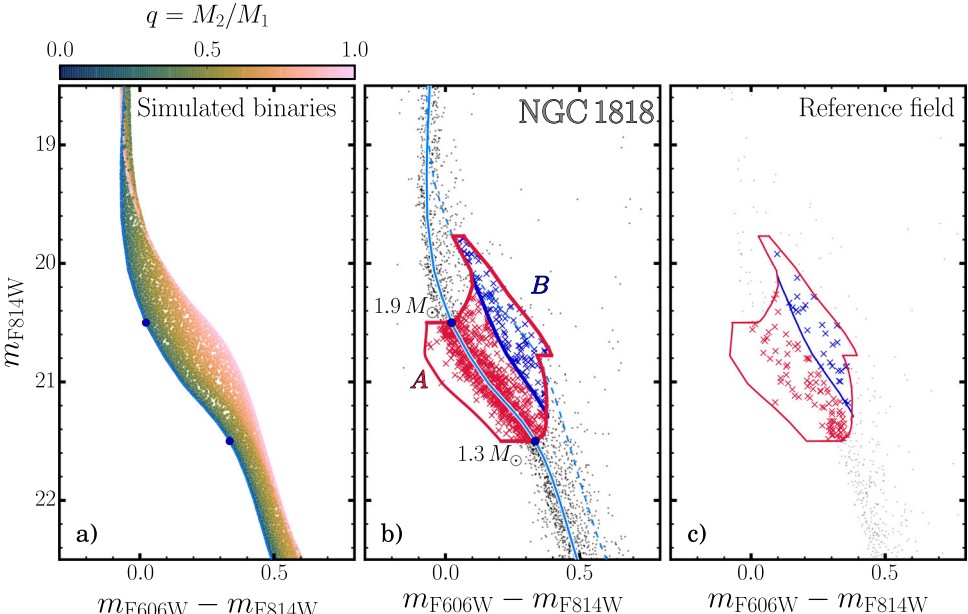

**Fig. 5 | Binary fraction of NGC 1818. a** CMD of a simulation of a population of 100,000 MS-MS binaries, with each binary system colour-coded according to its mass ratio $q = M_2/M_1$, where $M_2$ is the mass of the secondary star and $M_1$ the mass of the primary star, as shown in the top colourbar. **b** Unresolved binaries selection procedure for cluster-field stars. The light blue solid and dashed line represents the MS fiducial line and the MS-MS equal mass binaries fiducial line, respectively. The leftmost solid red line corresponds to the MS fiducial line, shifted by $-3\sigma$, while the rightmost solid red line is the MS-MS equal mass binaries fiducial line shifted by $+3\sigma$. Finally, the blue solid line indicates the fiducial line MS-MS binary stars with $q = 0.7$, used to select binary stars. The two black dots are the magnitude limits of the binary region, with their mass indicated on the left. **c** Same as (**b**), but for stars in the reference-field.

mag, and $E(B-V) = 0.08$ mag for the distance modulus and reddening, respectively, and we assumed a metallicity $Z = 0.006$[22].

As illustrated in Fig. 5a, binary systems where the primary stars are more massive than $1.9M_\odot$ and less massive than $1.3M_\odot$ are almost indistinguishable from single MS stars in the $m_{F814W}$ vs. $m_{F606W}-m_{F814W}$ CMD. On the contrary, in the interval between 1.3 and 1.9 solar masses, binary stars with mass ratio $q > 0.7$ are clearly separated from MS stars and populate a region of the CMD that is redder and brighter than the locus of single MS stars.

To estimate the fraction of MS-MS binaries we exploited the $m_{F814W}$ vs. $m_{F606W}-m_{F814W}$ CMD, where all MS stars distribute along a single sequence and follow the procedure introduced by Milone et al.[48] and extended to young open clusters by Cordoni et al.[6].

The main steps are illustrated in Fig. 5b, c, where we show the observed CMDs of stars in the cluster- and reference-field, respectively.

Region A of the CMD comprises all single stars with masses between 1.3 and 1.9 $M_\odot$ and all binary systems where the mass of the primary component lies in the same mass interval. Region B is the portion of region A, redder than the fiducial lines of binaries with mass ratio $q = 0.7$. Stars in region B are represented with blue crosses in Fig. 5b, c, while the remaining stars in region A are coloured in red.

In addition to cluster stars, regions A and B are also populated by field stars as shown in the CMD of reference-field stars (Fig. 5c).

The fraction of binaries with $q > 0.7$ is estimated as:

$$f_{\mathrm{bin}}^{q>0.7} = \frac{N_{cf}^B - N_{rf}^B}{N_{cf}^A - N_{rf}^A} - \frac{N_{AS}^B}{N_{AS}^A} \qquad (1)$$

where *cf*, *rf*, *AS* stand for cluster field, reference field and artificial stars, respectively. Artificial stars are used to estimate the fraction of single stars that, due to observational uncertainties, populate region B.

By assuming a flat mass-ratio distribution, as inferred for Galactic GCs and $q > 0.5$[48], we extrapolate a total fraction of binaries of $0.37 \pm 0.04$. Such value is consistent with the findings of Li et al.[49], in

the case of no cut-off and flat mass-ratio distribution (solid blue line in the bottom panel of their Figure 15).

### The luminosity function

To compute the LF of cluster stars shown in Fig. 1c, we determined the number of stars in different magnitude intervals, with a fixed width of 0.15 magnitudes. Such width has been chosen in order to have good statistics and sensitivity in each bin. To test the effect of different bin sizes we repeated the analysis for different bin widths from 0.13 to 0.17 mag, finding nearly-identical results. Incompleteness has been taken into account by weighting each star with the corresponding completeness level, derived for the cluster and reference field individually.

We applied the same procedure to both cluster- and reference-field stars, and we finally subtracted the latter to the former, by accounting for the different areas of the cluster and reference fields. This approach allows us to remove the contribution of field stars in the LF of cluster members. To test the robustness of the subtraction procedure, we verified that nearly identical results are obtained performing the field subtraction directly on the CMD, instead of on the LF.

Statistical uncertainties associated with each bin-count have been inferred by means of bootstrapping. We re-sampled with replacements 1000 times the observed CMD, and for each realization, we recomputed the LF. Uncertainties correspond to the dispersion of each bin-count distribution and are represented by the dark-grey shades in the LF plotted in Fig. 1d. The LF has been derived for stars brighter than $m_{F814W} = 25.5$, which is the luminosity interval where the completeness is higher than 50%.

### Age determination from the main-sequence turn-off

The hypothesis that age variations are responsible for the eMSTO would provide the opportunity of inferring the age distribution of NGC1818 stars.

To do this, we exploited the $m_{F814W}$ vs. $m_{F336W}-m_{F814W}$ CMD plotted in Fig. 2, which is the photometric diagram where the eMSTO is

more evident. We selected all eMSTO stars brighter than $m_{F814W} = 17.5$ and assumed that the colour and magnitude broadening of MSTO stars is entirely due to multiple stellar generations with different ages.

Following the recipe by Cordoni et al.[6], we created a grid of Padova isochrones[47] by assuming the same distance, reddening and metallicity derived above and ages between 10 and 110 Myr in steps of 2 Myr.

The age of each star has been determined by linearly interpolating the stellar colours over the grid of isochrones. The histogram distribution of the resulting ages is provided in Fig. 2c.

### Age determination from the main-sequence turn-on

The turn-on represents the point, along the zero-age main-sequence (ZAMS) where pre-main-sequence stars join the MS, and therefore translates into an over-density of stars detectable in the LF.

The turn-on comprises stars of different masses than those in the MSTO region and therefore represents an independent clock to determine the age of the cluster. Furthermore, the magnitude of the TOn changes with cluster age, moving to fainter magnitudes for older ages[28,29], so that the presence of multiple populations would result in the appearance of multiple or broad turn-on peaks in the cluster LF.

To illustrate the sensitivity of the turn-on on cluster age, we show in Fig. 3a the isochrones and LFs of four different stellar populations with ages of 10, 25, 50 and 80 Myr, indicated by different colours, together with the ZAMS, indicated by the dashed black line. Figure 3b–e shows how the luminosity of the turn-on, corresponding to the peak in the LF, changes by nearly four magnitudes in the considered age range, thus being clearly detectable considering the observational uncertainties of the present dataset. Moreover, MIST stellar models[38–40], available for two different stellar rotation rates (namely $\omega = 0$, $0.4 \, \omega_{crit}$, with $\omega_{crit}$ being the critical rotational velocity of the star), show that stellar rotation does not affect the position of stars in the CMD below the MS knee, i.e. the point where the MS bends due to the onset of surface turbulence, and therefore the position of the turn-on is poorly affected by stellar rotation.

Hence, if NGC 1818 is composed of a simple stellar population (SSP), its LF would show a single peak corresponding to the turn-on magnitude, while, if two or more stellar generations are present, the same number of peaks would appear in the LF. This approach represents an alternative and unbiased perspective to shed light on the eMSTO phenomenon in Magellanic Clouds clusters.

To infer the age of NGC 1818 we exploited a similar approach to that described in refs. 29–31, which is based on the comparison between the observed luminosity function and grids of synthetic CMDs generated from isochrones.

We performed a $\chi^2$ analysis, allowing for the presence of distinct stellar populations with ages in the range 10–100 Myr. The number, $N$, of stellar populations is determined by the adopted time duration of each star formation episode, and from the age range of the simulations. We adopted a duration of $\Delta t = 2.5$ Myr, which provides $N = 36$.

We first created a set of synthetic CMDs from the Padova isochrones[47], adopting a Kroupa[50] initial mass function, and a continuous star formation history lasting $\Delta t = 2.5$ Myr, so that each simulation contains stars with ages within $t_{min}$ and $t_{min} + \Delta t$. Each synthetic stellar population contains 200,000 stars. To test the influence of the adopted stellar models, we repeated the analysis exploiting MIST stellar models[38–40], obtaining consistent results. Specifically, we find a mean age of 40.2 Myr and an age spread of 6.0 Myr, determined as the FWHM of the best fit Gaussian (or 10.2 Myr when accounting for the full width of the Gaussian function).

We then artificially added a population of binary stars corresponding to the observed binary fraction and we convolved each simulation according to the observational uncertainties determined from artificial-star tests. Each simulated CMD is used to derive the corresponding LF.

Finally, we combined together the LFs of the simulated SSPs. Specifically, the LF of the population $j$ is multiplied by a scaling factor, $C_j$, which indicates the contribution of the LF of the population $j$ to the final combined LF. Therefore, $C_j$ is a value that can range from 0 to 1, where the former implies the absence of the corresponding population, i.e. no net contribution to the composite LF. The simulated LFs have been compared with the observed one and the best fit have been determined by minimizing the Poissonian $\chi^2$ [51] in Eq. 2

$$\chi^2 = \sum_i^{bins} n_i \ln(n_i/m_i) - n_i + m_i \tag{2}$$

where $n_i$ and $m_i$ are the bin values of the observed and simulated LFs, while the index $i$ index runs over the magnitudes bin. The $\chi^2$ has been derived in the magnitude interval between $m_{F814W} = 18.0$ and $m_{F814W} = 25.5$, which comprises non-saturated stars with completeness larger than 0.5.

To minimize the $\chi^2$, we used the genetic algorithm from the geneticalgorithm Python public library (https://pypi.org/project/geneticalgorithm/), which prevents us from finding a local minimum, instead of the global one.

As output, we retrieve an array of $N = 36$ coefficients $C_j$ that provides the contribution of each simulated stellar population to the best-fit LF. The age distribution of the stars that provide the best-fit LF is shown in Fig. 4c and consists of our best estimate for the age distribution in NGC 1818.

Uncertainties are inferred by bootstrapping the results 1000 times. Briefly, we sampled the observed CMD with replacements 1000 times, and for each sample we re-performed the fitting of the derived LF, thus obtaining a distribution of 1000 output coefficients $C_j$.

## Data availability

The authors declare that the data supporting the findings of this study are available within the paper and in Supplementary Data 1. The datasets generated during and/or analysed during the current study are available from the corresponding author on reasonable request. Source data are provided with this paper.

## Code availability

The computer programmes used to reduce HST images (i.e. KS2) are fully described and referenced in refs. 41–45. Parts of the Fortran routines are publicly available on Jay Anderson personal website https://www.stsci.edu/~jayander/. The complete code is available from the corresponding author upon reasonable request. Computer codes used to generate the results discussed in the present work are based on the procedure described in refs. 33, 34 and are adapted to the present work. Such code is available from the corresponding author upon reasonable request.

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

## Acknowledgements

This work has received funding from the European Research Council (ERC) under the European Union's Horizon 2020 research innovation programme (Grant Agreement ERC-StG 2016, No. 716082'GALFOR', PI: Milone, http://progetti.dfa.unipd.it/GALFOR). A.P.M., E.D. and

M.T. and G.C. acknowledge support from MIUR through the FARE projectR164RM93XW SEMPLICE (PI: Milone). A.P.M. and M.T. have been supported by MIUR under the PRIN programme 2017Z2HSMF (PI: Bedin). M.T. acknowledges support from the European Research Council Consolidator Grant funding scheme (project ASTEROCHRONOMETRY, G.A. n. 772293, http://www.asterochronometry.eu).

## Author contributions

G.C., A.P.M. and A.F.M. designed the study, coordinated the activity, and wrote the paper and the HST proposal GO-15495. The data have been reduced by G.C. and A.P.M., and the data analysis has been carried out by G.C. M.C. provided suggestions for the analysis of the MS turn-on. A.M., E.D., E.P.L., M.C., M.T., M.V.L. and S.J. contributed to the discussion and commented upon the manuscript.

## Competing interests

The authors declare no competing interests.
