## [Peer Review File · Nature Communications]

REVIEWER COMMENTS

Reviewer #1 (Remarks to the Author):

2nd Review of Cordoni et al., The Turn-On of NGC1818 unveils the origin of Multiple Stellar Populations in Magellanic Cloud clusters; *Nature Communications*, re-submitted

I thank the authors for considering the issues I raised in my first report (Referee #1). I am mostly happy with the revised version of this paper, and I have no objections to it being published in *Nature Communications*.

Before recommending acceptance, however, I would like the authors to consider the following comments, provided as constructive criticism to help them further improve their paper.

1. There are still some instances of imprecise language. This is particularly obvious in the first, bold-face paragraph. That paragraph, although not formally speaking an abstract, will be read first by most people, so it should be self-explanatory and clear. In fact, the burden on the authors to make this as clear as possible is greatest for this paragraph. Here are some suggestions for further clarifications:

1a. (blue text) "... along the colour-magnitude diagrams ..." is imprecise. The multiple sequences are found alongside SPECIFIC FEATURES in CMDs. "Along CMDs" doesn't make sense here, I'm afraid.

1b. "... the Turn-Off of the [CMD] ..." This is also imprecise. It's not the turn-off of the CMD, but the turn-off of the main sequence in the CMD...

1c. "... of NGC1818 ..." This comes out of the blue. What is NGC 1818, why is this an interesting object, etc. In the bold-face paragraph, you must make sure that the reader understand what the

science is about, without having to go down a few pages to find out that the object is a young star cluster in the LMC (you should probably also include its age here).

2. On p. 2, where AGB stars are mentioned in association with a 50-150 Myr age spread, I would like to see references supporting those numbers; also, I was under the impression that AGB-induced age spreads are often suggested to encompass 300 Myr or even more, so why limit this to up to 150 Myr?

3. p. 2, penultimate para (The eMSTOs have ..."): the authors argue that young MC clusters may be counterparts of old GCs. However, that is too simplistic. The peak of the GC mass function is at $2 \times 10^5 M_{\text{sun}}$, with very few clusters at masses of a few $\times 10^4 M_{\text{sun}}$, which is where these MC clusters come in. In addition, since these MC clusters are young, standard stellar evolution will make them lose at least 15% of their mass, if not more, over the course of their evolution to ages of 10-12 Gyr. Plus many will be dissolved dynamically given their low masses. As such, the chances that these MC clusters survive to become old GCs are really small. And even if they survive, their metallicities will be much higher than those of the current population of old GCs. These arguments taken together suggest to me that the young populous MC clusters are not really good counterparts to old Galactic GCs, even for comparison purposes...

4. p. 3, top para: I agree that rotation on its own may not be sufficient to explain the extents of the eMSTOs. However, why do the authors categorically state that the remainder of the extents must come from age differences? What about metallicity differences or differences in the light-element abundances?

4a. The same comment applies to the first para in the section titled "The nature of the eMSTO" on p. 5.

5. Throughout: I was a bit puzzled by the authors' use of the term "azure" for blue. While not technically incorrect, readers may not immediately know that meaning, so I recommend that they use "blue" instead.

6. In the caption of Fig. 2, I recommend that the authors add the step size between subsequent non-rotating isochrones in the top left panel.

Reviewer #2 (Remarks to the Author):

Report for the paper “The Turn-On of NGC1818 unveils the origin of Multiple Stellar Populations in Magellanic Cloud clusters” by Cordoni et al.

The results reported in this paper are very interesting and extremely important in the context of understanding the phenomenon of the extended main sequence turn off (eMSTO) and split main sequence (MS) in star clusters younger than 2 Gyr.

I also believe that the analysis is robust and the technique used in the paper is well exploited for its purpose.

However, I have two major comments about the general discussion of the results and the context in which the paper was framed:

1) Throughout the paper, the authors claim that the reported results unveil the origin of multiple stellar populations, by connecting them with the multiple populations phenomenon observed in the ancient globular clusters (GCs).

Multiple populations in old GCs manifest in the form of chemical abundance variations (He, Na, C, N etc), while in young star clusters no such chemical variations are observed to date (the youngest cluster where these are found in the form of N spread is NGC 1783, ~1.7 Gyr old, see the recent paper from Cadelano et al. 2022, ApJ, 924L, 2C).

It is true and well established that young star clusters show extended main sequence turn offs and split main sequences in their CMDs that are definitely not “simple stellar populations”. Nevertheless, even if these phenomena and the multiple populations in old GCs might be related, a connection is not established, to date. Indeed, no spectroscopic measurements of chemical abundance variations along the eMSTO or split MS of young star clusters exist to prove this. Additionally, Milone et al. (2015, MNRAS, 450, 3750; 2016, MNRAS, 458, 4368) showed that stellar isochrones including chemical abundance spreads are not able to reproduce the observed bi-modal main sequences in young Magellanic Cloud clusters. A quote from their paper is “In contrast, isochrones with different helium abundance do not reproduce the observed CMD of NGC 1755. This fact provides a significant difference between the multiple sequences observed in the old Galactic GCs and in the young MC [Magellanic Cloud] clusters.”

In this context, I believe that the paper needs to be extensively re-written in order to connect the importance of its findings to a less generalised topic. The result of this paper is definitely a breakthrough into the debate between stellar rotation and age spreads to explain the phenomenon of the eMSTO and split MSs and it deserves publication. However, I found the connection with the multiple population phenomenon in the old GCs to be not relevant and, at the same time, confusing for a non-expert reader.

2) Despite being certainly a new and exciting result based on a novel technique, I found this paper to lack a few key references of works in which other authors attempted to find or disprove the presence of age spreads within young star clusters.

For example, Cabrera-Ziri et al. (2014, MNRAS, 441, 2754C), (2015, MNRAS, 448, 2224C), (2016, 457, 809C) found no evidence of extended star formation history in young star clusters in starburst galaxies, as well as Hollyhead et al. (2015, MNRAS, 449, 1106H).

A similar result to Cordoni et al. (2018, ApJ, 869, 139C) (cited in this paper), was found by Piatti & Bastian (2016, MNRAS, 463, 1632P) and Bastian et al. (2018, MNRAS, 480, 3739B) where they show that clusters that are not so massive (down to a few times $\sim 10^3 M_{\text{sun}}$) also have eMSTOs. According to the age spread scenario for the origin of the eMSTO, these low mass clusters should not have been able to retain enough stellar ejecta to go through a prolonged period of star formation.

Furthermore, a key demonstration that extended star formation might be unlikely in clusters with an eMSTO was provided by the $\Delta(\text{Age})$ vs Age relation, from the work by Niederhofer et al. (2015, MNRAS, 453, 2070N). Such a relation showed that the older the cluster is, the larger the eMSTO width. Hence, this points towards a stellar evolutionary effect, most likely stellar rotation (which was nicely confirmed by Marino et al. 2018, ApJL, 863, L33, cited already in this paper).

Another relevant paper is the one by Gossage et al. (2019, ApJ, 887, 199G), where the authors provide quantitative assessments of the eMSTO morphology of several young clusters by modelling the effect of age spread, stellar rotation and both at the same time. They found that “a distribution of rotation rates appears to be the overall most physically motivated explanation for the eMSTO phenomenon”. The results from this paper are in agreement with what found by Gossage+19 and collaborators.

Hence, I suggest the authors to remodel the introduction as mentioned above in point 1), by also adding and discussing most of the references reported in point 2).

Below I report some additional comments throughout the text of the paper:

-first two paragraphs of the introduction:

As explained in the major points above, I suggest then to remove these paragraphs that connect old GCs and young star clusters and instead focus on the phenomena of eMSTO and split MS in young star clusters (<2 Gyr old).

More generally, I also believe that a comparison with old GCs can be made throughout the text with a few sentences, as it is already done when stating that an estimation of age spreads in ancient GCs is hard because of stellar evolution (page 2, third paragraph).

-Introduction last paragraph: sentence "Hence, the debate is still entertained on whether the young clusters host multiple stellar generations or if the eMSTO and the multiple populations of old GCs are different phenomena".

This is true but it is not relevant to the current study (see points above). The multiple stellar populations present in ancient GCs are star-to-star chemical abundance variations. We would need spectroscopic studies of stars along the eMSTO or split MSs to prove that also these phenomena show chemical abundance variations, at least.

-General comment about the nomenclature "multiple generations":

Throughout the text, the authors often use "multiple generations" to refer to the multiple stellar populations, both for young and ancient clusters. This implies that a second (or third etc...) generation formed from the ejecta of a first generation of (massive) stars within the same cluster. Although being one of the possible scenarios envisioned for the origin of the chemical abundance variations in old GCs, this model suffers from many unresolved issues, as well as other scenarios do. It is not established or proved that the multiple populations in old GCs are formed through this channel. Hence, I would prefer the authors to use more "neutral" terms such as "multiple stellar populations", if they refer to ancient GCs.

Secondly, "generation" or "multiple populations" should not be used to describe the populations present in NGC 1818 (such as the eMSTO and split MS) because it infers as well that the same type of populations are observed in eMSTO/split MS and in ancient GCs. I suggest to use "different populations" when referring to NGC 1818 or "the populations", for instance.

-page 2 paragraph named "NGC 1818. The 'Rosetta stone' to disentangle age and rotation."

The authors state here that the "main properties of the multiple populations in young star clusters do not depend on cluster mass". However, above, in page 2 paragraph 5, they stated that "the age spread inferred from the eMSTO correlated with cluster mass, in close analogy with what is observed in old GCs, where the complexity of multiple populations increases with the mass of the host cluster".

Which main properties of young star clusters the authors referred to in the first sentence?

Also, the second sentence seems to contradict the first one.

Could you please elaborate more what you mean by these two sentences?

-page 5, paragraph "The nature of the MSTO"

From "In addition to [...] disentangle the two scenarios".

I believe that this sentence is very confusing and I strongly suggest to remove it, along with the references. The studies mentioned in the references (numbers 50, 51, 52) looked at multiple populations in intermediate age/young star clusters as chemical abundance variations and are most likely not linked to the phenomenon of the eMSTO, neither to stellar rotation.

-In Methods:

In Cignoni et al. (2010, ApJL, 712, L63), the authors list the sources of uncertainties arising when calculating the age spread from the Turn-On feature of young star clusters. The majority of these uncertainties were considered in the current paper, however, has the differential reddening being checked for NGC 1818? I believe it should not significantly affect the result of the paper, but a few sentences about it would be useful in the Methods section.

-Luminosity function paragraph in Methods:

The authors write about the procedure of subtracting the reference field stars to the cluster stars. How are they sure that they are not subtracting cluster stars and hence contaminating the results?

It would be nice to show here as reply to the report (probably there is no need in the paper) a map of the field of view of the observations with the selected regions for both cluster and reference stars, as well as the CMDs of the cluster and reference stars.

-Figure 1:

On the x axis of the right panel the authors label "Normalised Counts". Does this represent the luminosity function? If not, how are these normalised counts calculated? (the same applies to Figure 3). In order to clarify, I would suggest either to change the label to "Luminosity function(Counts)" or, if different, please explain in the text what "Normalised" means.

Also, does the CMD of the left panel of Figure 1 represent the field-stars cleaned CMD? By comparing the histogram with the left panel of Figure 1 (binned CMD), it is possible to notice that around F814W mag of 21.8-22, the star counts are generally higher (black pixels) with respect to the Turn On peak (dark red pixels). Is this due to the fact that the width of the MS around the TOn peak is larger?

Generally, it would be useful to (i) add a colour bar of the star counts to the left panel, (ii) consistently plot the left and right panels, to show the field star subtracted CMD and counts.

-Figure 3:

I suggest to add a legend in the top two panels referring to the red and blue histograms.

Additionally, the same comment of the point above is valid for the simulated histograms. What does “normalised” mean in this figure? Were the simulations calculated with a percentage of field stars or in this case field stars were not considered? In any case, I suggest to expand about it in the Method section.

-Figure A3:

This figure is wrongly labelled as Figure A2, while it should instead be A3. Also, it is not clear what the black dashed line refers to? It is not shown in the legend.

If it represents NGC 1818 observation ridge line, why this is not consistent with a ~40 Myr old population?

AUTHORS:

We thank the anonymous referees for their helpful comments and suggestions that have improved the quality of the manuscript. In the following, we answer each point individually, and we have modified the main text as discussed below (all changes in the main paper are marked with blue text).

REVIEWER COMMENTS

Reviewer #1 (Remarks to the Author):

2nd Review of Cordoní et al., The Turn-On of NGC1818 unveils the origin of Multiple Stellar Populations in Magellanic Cloud clusters; Nature Communications, re-submitted

I thank the authors for considering the issues I raised in my first report (Referee #1). I am mostly happy with the revised version of this paper, and I have no objections to it being published in Nature Communications.

Before recommending acceptance, however, I would like the authors to consider the following comments, provided as constructive criticism to help them further improve their paper.

1. There are still some instances of imprecise language. This is particularly obvious in the first, bold-face paragraph. That paragraph, although not formally speaking an abstract, will be read first by most people, so it should be self-explanatory and clear. In fact, the burden on the authors to make this as clear as possible is greatest for this paragraph. Here are some suggestions for further clarifications:

1a. (blue text) "... along the colour-magnitude diagrams ..." is imprecise. The multiple sequences are found alongside SPECIFIC FEATURES in CMDs. "Along CMDs" doesn't make sense here, I'm afraid.

1b. "... the Turn-Off of the [CMD] ..." This is also imprecise. It's not the turn-off of the CMD, but the turn-off of the main sequence in the CMD...

1c. "... of NGC1818 ..." This comes out of the blue. What is NGC 1818, why is this an interesting object, etc. In the bold-face paragraph, you must make sure that the reader understand what the science is about, without having to go down a few pages to find out that the object is a young star cluster in the LMC (you should probably also include its age here).

AUTHORS:

We clarified the text following the reviewer's comments

2. On p. 2, where AGB stars are mentioned in association with a 50-150 Myr age spread, I would like to see references supporting those numbers; also, I was under the impression that AGB-induced age spreads are often suggested to encompass 300 Myr or even more, so why limit this to up to 150 Myr?

AUTHORS:

The upper limit of 150 Myr is imposed by the fact that low-mass AGB stars do not produce the right chemical composition for reproducing the 2G in mono-metallic GCs. We added references in the paper, while in the following we provide further details on why the AGB scenario excludes long star-formation durations.

Indeed, the classical AGB scenario suggests that only intermediate-mass AGB stars (i.e. 4 - 8 solar masses) are responsible for the formation of 2G stars. Following the detailed discussion in D'Ercole et al. (2010, MNRAS, 407, 854), the limiting age ranges from ~ 100 Myr (if second generations form from pure AGB ejecta) to ~ 150 Myr (in the case of dilution with pristine gas). Subsequent versions of the AGB scenario included super-AGB stars as additional polluters so that the lower limit to the age spread is about 50 Myr (e.g. Bastian & Lardo 2018 and references therein).

While the exact mass range is somehow model-dependent (e.g. Renzini et al. 2015, D'Ercole et al. 2010 for the caveats of their models), it is widely accepted that stars less massive than 3-4 solar masses can not be responsible for the formation of 2G stars.

Such constraint is due to the fact that the yields of low-mass AGB stars are strongly enhanced in the overall C+N+O abundance and in s-process elements, in contrast with what is observed in most GCs, where the abundances of these elements are constant.

It should be noticed that in the case of the NGC1851, which is an anomalous GCs with internal variation in metallicity, s-process elements (Yong et al. 2008) and, possibly, in C+N+O (Yong et al. 2009, but see Tautvaišienė et al. 2022) low mass AGB stars have been invoked to explain the presence of the anomalous population. Similar arguments have been applied to the GCs M22 and omega Cen (e.g. Marino et al. 2012). In these cases, the time interval for the formation of the anomalous population would exceed 150 Myr, and supernovae, together with AGB, play a role in the SFH of the clusters.

In this work, we would prefer not to discuss the specific cases of GCs with heavy-element variations and refer to the classic AGB scenarios.

3. p. 2, penultimate para (The eMSTOs have ...): the authors argue that young MC clusters may be counterparts of old GCs. However, that is too simplistic. The peak of the GC mass function is at $2 \times 10^5 M_{\text{sun}}$, with very few clusters at masses of a few $\times 10^4 M_{\text{sun}}$, which is where these MC clusters come in. In addition, since these MC clusters are young, standard stellar evolution will make them lose at least 15% of their mass, if not more, over the course of their evolution to ages of 10-12 Gyr. Plus many will be dissolved dynamically given their low masses. As such, the chances that these MC clusters survive to become old GCs are really small. And even if they survive, their metallicities will be much higher than those of the current population of old GCs. These arguments taken together suggest to me that the young populous MC clusters are not really good counterparts to old Galactic GCs, even for comparison purposes...

AUTHORS:

We agree that young MC clusters and old Galactic GCs have different physical properties and would follow different dynamical evolution. Following the suggestions of the referee, we revised the text on comparison between young MC clusters and old Galactic GCs.

In particular, we deleted the sentence that MC could be the counterparts of GCs, which was misleading/incorrect, by quoting Conroy et al. (2011) and Keller et al. (2011), who suggested that a similar mechanism is responsible for the formation of multiple populations in eMSTO clusters and in old GCs .

Moreover, even though young MC clusters and old Galactic GCs exhibit different physical properties, such as mass and metallicity, we believe that the connection between these two objects, and their multiple populations, is worth exploring. Specifically, we refer to the points discussed later in the present report, e.g. point 1 on pages 4 and 5.

4. p. 3, top para: I agree that rotation on its own may not be sufficient to explain the extents of the eMSTOs. However, why do the authors categorically state that the remainder of the extents must come from age differences? What about metallicity differences or differences in the light-element abundances?

4a. The same comment applies to the first para in the section titled "The nature of the eMSTO" on p. 5.

AUTHORS:

Following Milone et al. (2016, e.g. their Fig. 9a-b), we exclude helium and iron variations as responsible for the eMSTO, because stellar populations with different Z or Y would not reproduce the observed eMSTO, MSs and RGBs.

As an example, stellar populations with different helium abundances would result in split or broad RGB sequences, in contrast with the observations. Similarly, the merging of the MSs in young clusters is not consistent with significant differences in metals and helium.

We also exclude that the broadening/split of the eMSTO and bright MS stars are due to star-to-star variations in other light elements (such as C, N and O). Indeed, any variation in light elements, if present, would not provide a detectable change in the flux of these stars. Moreover, these elements would not affect the stellar fluxes in optical filters even in colder stars.

Because of these reasons, we assume that the only mechanism able to explain the residual spread in the Turn-Off region must be the age difference. We updated the main text clarifying this point.

5. Throughout: I was a bit puzzled by the authors' use of the term "azure" for blue. While not technically incorrect, readers may not immediately know that meaning, so I recommend that they use "blue" instead.

AUTHORS:

We changed 'azure' with 'blue'.

6. In the caption of Fig. 2, I recommend that the authors add the step size between subsequent non-rotating isochrones in the top left panel.

AUTHORS:

We added the step size in the caption of Fig. 2

Reviewer #2 (Remarks to the Author):

Report for the paper "The Turn-On of NGC1818 unveils the origin of Multiple Stellar Populations in Magellanic Cloud clusters" by Cordoni et al.

The results reported in this paper are very interesting and extremely important in the context of understanding the phenomenon of the extended main sequence turn off (eMSTO) and split main sequence (MS) in star clusters younger than 2 Gyr.

I also believe that the analysis is robust and the technique used in the paper is well exploited for its purpose.

However, I have two major comments about the general discussion of the results and the context in which the paper was framed:

1) Throughout the paper, the authors claim that the reported results unveil the origin of multiple stellar populations, by connecting them with the multiple populations phenomenon observed in the ancient globular clusters (GCs).

Multiple populations in old GCs manifest in the form of chemical abundance variations (He, Na, C, N etc), while in young star clusters no such chemical variations are observed to date (the youngest cluster where these are found in the form of N spread is NGC 1783, ~1.7 Gyr old, see the recent paper from Cadelano et al. 2022, ApJ, 924L, 2C).

It is true and well established that young star clusters show extended main sequence turn offs and split main sequences in their CMDs that are definitely not “simple stellar populations”.

Nevertheless, even if these phenomena and the multiple populations in old GCs might be related, a connection is not established, to date. Indeed, no spectroscopic measurements of chemical abundance variations along the eMSTO or split MS of young star clusters exist to prove this.

Additionally, Milone et al. (2015, MNRAS, 450, 3750; 2016, MNRAS, 458, 4368) showed that stellar isochrones including chemical abundance spreads are not able to reproduce the observed bi-modal main sequences in young Magellanic Cloud clusters. A quote from their paper is “In contrast, isochrones with different helium abundance do not reproduce the observed CMD of NGC 1755. This fact provides a significant difference between the multiple sequences observed in the old Galactic GCs and in the young MC [Magellanic Cloud] clusters.”

In this context, I believe that the paper needs to be extensively re-written in order to connect the importance of its findings to a less generalised topic. The result of this paper is definitely a breakthrough into the debate between stellar rotation and age spreads to explain the phenomenon of the eMSTO and split MSs and it deserves publication. However, I found the connection with the multiple population phenomenon in the old GCs to be not relevant and, at the same time, confusing for a non-expert reader.

AUTHORS:

As suggested by the referee, we modified some text in the paper, including the abstract and the Introduction. We also clarify that our paper does not unveil the origin of multiple populations in GCs.

Nevertheless, we would prefer to keep some discussion on multiple populations in old GCs, and the conclusion that “multiple populations in GCs either coeval or formed with different mechanisms than the multiple populations of young clusters.”

Indeed, we believe that the notion of whether multiple populations in young and old clusters are similar phenomena or not is still an open and quite debated issue, and our work provides further constraints to the phenomenon.

In the following, we provide various arguments supporting our statement.

– **Chemical abundance variations** Chemical variations are one of the distinctive features of multiple populations in GCs and in several clusters older than about 1.5 Gyr.

The maximum internal variation in light elements in GCs correlates with cluster mass so that we expect that light element variations in low-mass clusters, if present, could be below the present detection threshold. We refer to Goudfrooij et al. (2014) for theoretical discussion.

This idea seems supported by the recent findings of multiple populations with different nitrogen abundance in the 1.5 Gyr old clusters NGC2173 (Kapse et al. 2022) and NGC1783 (Caledano et al. 2022). Here, the photometric detection of very-small nitrogen spread was allowed by the exquisite dataset, which had unprecedented quality in multiple-population studies of young clusters.

– **Age spreads** As mentioned now in the introduction, the small light-element variation, or even the lack thereof, could be consistent with a similar formation scenario in old and young clusters. If young clusters experienced prolonged star formation, it is reasonable to assume that clusters with chemical anomalies also host stellar populations of different ages. As an example, the recent discoveries of chemical anomalies in the eMSTO clusters NGC1783 and NGC2173 would fit in a scenario, where these clusters have experienced prolonged star formation and their N-rich stars are second-generation stars mostly from pristine material plus a small amount of ejecta from more massive 1G stars.

In general, scenarios for the formation of multiple populations in GCs, like the AGB scenario, suggest that 2G stars form from the mixing of pure ejecta and pristine material. The more pristine gas is accreted, i.e. more dilution, the more the chemical composition of 2G stars approaches that of 1G stars. The exact amount of pure ejecta and pristine material is poorly constrained and would depend on various quantities, including the mass of the polluter and the distribution of the pristine material.

Anyhow, in order to reproduce the abundance pattern observed in small-mass Galactic GC M4, where the internal sodium variation corresponds to ~ 0.4 dex and the mass fraction helium variation is 0.01 (e.g. Marino et al. 2008, Milone et al. 2018), most of the material from which 2G formed should be composed of pristine gas. By extending this scenario to the 2.5 Gyr old LMC cluster NGC1978, where the sodium variation between 1G and 2G stars is just 0.05 dex and there is virtually no helium variation (Saracino et al. 2020, Milone et al. 2020), we would conclude that only a tiny amount of AGB ejecta has been retained by the 2G star-forming region.

In this context, it would be reasonable to assume that a small amount of AGB ejecta has mixed with pristine material of the surrounding star-forming region, despite the smaller mass of this cluster.

By extending this argument to NGC1818, the ejecta could have been entirely, or widely, lost and the 2G would have formed from pristine material, such that the resulting abundance variations would be too small to be detected (we stress that with the available dataset, it would be not possible to detect in NGC1818 nitrogen variations that are as small as those observed in NGC1783 by Caledano et al. 2022). Hence, regardless of whether NGC1818 has retained AGB ejecta or not, it is crucial to constrain the duration of the star formation.

– **Stellar Mass** An additional reason why chemical anomalies are rarely detected in MC clusters younger than 2Gyr could be that most of the studies are based on RGB stars, which are more massive than about 1.5 solar mass. The AGB scenario by D’Ercole et al. (2008, 2010) assumes that multiple populations with different chemical composition only form below a certain mass threshold. Similarly, Bastian & Lardo (2018) suggested that chemical anomalies may originate among stars smaller than about 1.5 solar masses and are associated with the stellar structure.

The conclusion by Caledano et al. (2022) that chemical anomalies of NGC1783 are detected along the cluster MS (where stars have masses lower than 1.5 solar masses) but seem to be not present along the RGB is consistent with these predictions.

In this scenario, our result that 1 solar mass TOn stars of NGC1818 are nearly coeval would either challenge the AGB scenario or indicate that multiple populations in young clusters and in NGC1783 formed with different mechanisms.

– **Rotation** Milone et al. 2015, 2016 provided evidence of split MSs and eMSTO in young LMC clusters. Moreover, the 2016 paper on NGC1755 also shows that the split MS is not consistent with stellar populations with (large) variations in chemical composition but is well reproduced by isochrones with different rotation rates.

Nevertheless, a more accurate comparison between the observed and the simulated CMDs composed of stars with different rotation rates has revealed that the latter are not able to entirely reproduce the eMSTOs and that age difference, together with rotation, is needed to match the observation (see for e.g. Figure 10b in Milone et al. 2016, Figure 11 in Milone et al. 2017, Gossage et al. 2019 or the Goodfroom et al. 2017 paper ‘stellar rotation diminishes but does not eliminate age spread’).

2) Despite being certainly a new and exciting result based on a novel technique, I found this paper to lack a few key references of works in which other authors attempted to find or disprove the presence of age spreads within young star clusters.

For example, Cabrera-Ziri et al. (2014, MNRAS, 441, 2754C), (2015, MNRAS, 448, 2224C), (2016, 457, 809C) found no evidence of extended star formation history in young star clusters in starburst galaxies, as well as Hollyhead et al. (2015, MNRAS, 449, 1106H).

A similar result to Cordoni et al. (2018, ApJ, 869, 139C) (cited in this paper), was found by Piatti & Bastian (2016, MNRAS, 463, 1632P) and Bastian et al. (2018, MNRAS, 480, 3739B) where they show that clusters that are not so massive (down to a few times $\sim 10^3$ Msun) also have eMSTOs. According to the age spread scenario for the origin of the eMSTO, these low mass clusters should not have been able to retain enough stellar ejecta to go through a prolonged period of star formation.

Furthermore, a key demonstration that extended star formation might be unlikely in clusters with an eMSTO was provided by the $\Delta(\text{Age})$ vs Age relation, from the work by Niederhofer et al. (2015, MNRAS, 453, 2070N). Such a relation showed that the older the cluster is, the larger the eMSTO width. Hence, this points towards a stellar evolutionary effect, most likely stellar rotation (which was nicely confirmed by Marino et al. 2018, ApJL, 863, L33, cited already in this paper). Another relevant paper is the one by Gossage et al. (2019, ApJ, 887, 199G), where the authors provide quantitative assessments of the eMSTO morphology of several young clusters by modelling the effect of age spread, stellar rotation and both at the same time. They found that “a distribution of rotation rates appears to be the overall most physically motivated explanation for the eMSTO phenomenon”. The results from this paper are in agreement with what found by Gossage+19 and collaborators.

Hence, I suggest the authors to remodel the introduction as mentioned above in point 1), by also adding and discussing most of the references reported in point 2).

AUTHORS:

We modified the text as suggested and discussed all the results above.

As discussed in the paper, we agree that stellar rotation plays a major role in shaping the eMSTOs. Nevertheless, several papers conclude that rotation alone is not able to entirely reproduce the observed eMSTOs and some age spread is needed to match the observations (see Goudfrooij et al. 2017, Milone et al. 2017, Costa et al. 2018 and references therein). Even the detailed analysis by Gossage et al. 2019, ApJ 887 199 shows that while a distribution of rotation rates can reproduce the overall morphology of the eMSTO, a mixture of rotation rates and age spread is able to quantitatively match the observations.

It has been suggested that the inferred age spread is real (see papers by Goudfrooij and collaborators), or it is an artifact due to limitations in the available rotational models (e.g. Gossage et al. 2019) or it is due to phenomena associated with stellar breaking (e.g. D'Antona et al. 2018).

In any case, it is very challenging to disentangle among these possibilities by using the eMSTO alone. In contrast, the analysis of both the eMSTO and the MSTO introduced in this paper has the potential to address the question of whether eMSTO clusters host multiple stellar generations or not.

Below I report some additional comments throughout the text of the paper:

3A) first two paragraphs of the introduction:

As explained in the major points above, I suggest then to remove these paragraphs that connect old GCs and young star clusters and instead focus on the phenomena of eMSTO and split MS in young star clusters (<2 Gyr old).

More generally, I also believe that a comparison with old GCs can be made throughout the text with a few sentences, as it is already done when stating that an estimation of age spreads in ancient GCs is hard because of stellar evolution (page 2, third paragraph).

3B) Introduction last paragraph: sentence “Hence, the debate is still entertained on whether the young clusters host multiple stellar generations or if the eMSTO and the multiple populations of old GCs are different phenomena”.

This is true but it is not relevant to the current study (see points above). The multiple stellar populations present in ancient GCs are star-to-star chemical abundance variations. We would need spectroscopic studies of stars along the eMSTO or split MSs to prove that also these phenomena show chemical abundance variations, at least.

AUTHORS:

See our reply to point 1 for discussion of the two points above.

-General comment about the nomenclature “multiple generations”:

Throughout the text, the authors often use “multiple generations” to refer to the multiple stellar populations, both for young and ancient clusters. This implies that a second (or third etc...) generation formed from the ejecta of a first generation of (massive) stars within the same cluster. Although being one of the possible scenario envisioned for the origin of the chemical abundance variations in old GCs, this model suffers from many unresolved issues, as well as other scenarios do. It is not established or proved that the multiple populations in old GCs are formed through this channel. Hence, I would prefer the authors to use more “neutral” terms such as “multiple stellar populations”, if they refer to ancient GCs.

Secondly, “generation” or “multiple populations” should not be used to describe the populations present in NGC 1818 (such as the eMSTO and split MS) because it infers as well that the same type

of populations are observed in eMSTO/split MS and in ancient GCs. I suggest to use “different populations” when referring to NGC 1818 or “the populations”, for instance.

AUTHORS:

We modified the wordings as suggested by the referee, changing ‘Multiple Generations’ with ‘Multiple Populations’ whenever needed.

-page 2 paragraph named “NGC 1818. The ‘Rosetta stone’ to disentangle age and rotation.” The authors state here that the “main properties of the multiple populations in young star clusters do not depend on cluster mass”. However, above, in page 2 paragraph 5, they stated that “the age spread inferred from the eMSTO correlated with cluster mass, in close analogy with what is observed in old GCs, where the complexity of multiple populations increases with the mass of the host cluster”.

Which main properties of young star clusters the authors referred to in the first sentence?

Also, the second sentence seems to contradict the first one.

Could you please elaborate more what you mean by these two sentences?

AUTHORS:

We clarified these points in the text, as suggested.

-page 5, paragraph “The nature of the MSTO”

From “In addition to [...] disentangle the two scenarios”.

I believe that this sentence is very confusing and I strongly suggest to remove it, along with the references. The studies mentioned in the references (numbers 50, 51, 52) looked at multiple populations in intermediate age/young star clusters as chemical abundance variations and are most likely not linked to the phenomenon of the eMSTO, neither to stellar rotation.

AUTHORS:

We removed the sentence as suggested.

-In Methods:

In Cignoni et al. (2010, ApJL, 712, L63), the authors list the sources of uncertainties arising when calculating the age spread from the Turn-On feature of young star clusters. The majority of these uncertainties were considered in the current paper, however, has the differential reddening being checked for NGC 1818? I believe it should not significantly affect the result of the paper, but a few sentences about it would be useful in the Methods section.

AUTHORS:

We verified that NGC1818 is not affected by significant differential reddening, i.e. reddening variation, if present, is smaller than $E(B-V)=0.003$ mag, and is much smaller than photometric errors. We added a sentence in the Methods section.

-Luminosity function paragraph in Methods:

The authors write about the procedure of subtracting the reference field stars to the cluster stars.

How are they sure that they are not subtracting cluster stars and hence contaminating the results?

It would be nice to show here as reply to the report (probably there is no need in the paper) a map of the field of view of the observations with the selected regions for both cluster and reference stars, as well as the CMDs of the cluster and reference stars.

AUTHORS:

As discussed in the Methods section, to subtract the contribution of field stars we selected a region with radial distance from the cluster center larger than 75 arcsec, where can fairly assume that there are no cluster stars. As suggested, the cluster- and reference-field regions are shown in the top panel of Figure 2 of the present report, defined as the region inside and outside the red and blue circles, respectively.

Moreover, to investigate the effect of the field subtraction procedure, we repeated the analysis for different choices of the cluster field.

Figure 1 of this report illustrates the results for different cluster field radii, from 40 to 52 arcsec, and shows that the conclusions of the paper are not affected by this choice.

-Figure 1:

On the x axis of the right panel the authors label “Normalised Counts”. Does this represent the luminosity function? If not, how are these normalised counts calculated? (the same applies to Figure 3). In order to clarify, I would suggest either to change the label to “Luminosity functon(Counts)” or, if different, please explain in the text what “Normalised” means.

Also, does the CMD of the left panel of Figure 1 represent the field-stars cleaned CMD?

By comparing the histogram with the left panel of Figure 1 (binned CMD), it is possible to notice that around F814W mag of 21.8-22, the star counts are generally higher (black pixels) with respect to the Turn On peak (dark red pixels). Is this due to the fact that the width of the MS around the TOn peak is larger?

Generally, it would be useful to (i) add a colour bar of the star counts to the left panel, (ii) consistently plot the left and right panels, to show the field star subtracted CMD and counts.

AUTHORS:

As discussed below, we have modified Figure 1 of the manuscript including the bottom panels of Figure 2 of the present report. In the new version of the Figure, we did not normalize the histogram. Hence, the comment of the referee refers now to Figure 3 only (in the manuscript).

We clarify that stellar counts are normalised in such a way that the Turn-On peak corresponds to one. This choice is mostly for illustration purposes.

The CMD shown in Figure 1 includes all the stars in the cluster field, without statistically subtracting field stars.

We emphasise here that we preferred not to statistically subtract field stars from the cluster-field CMD, because we accounted for field-star contamination when we calculated the LF of cluster stars. To do that, we subtracted the completeness-correct LF of field stars from the completeness-correct cluster-field LF (see Section Method for details). We clarified the text and the Figure as suggested. We confirm that the different pixel colour is due to the different MS width.

We also modified the CMD of Figure 1 to illustrate the contribution from field stars to the result, and, as suggested by the referee, we added the colourbar. In the rightmost panel, we show the completeness corrected and field subtracted Luminosity Function (black lines), together with the reference field stars and cluster field stars LF, indicated by the blue and red lines, respectively.

-Figure 3:

I suggest to add a legend in the top two panels referring to the red and blue histograms. Additionally, the same comment of the point above is valid for the simulated histograms. What does “normalised” mean in this figure? Were the simulations calculated with a percentage of field stars or in this case field stars were not considered? In any case, I suggest to expand about it in the Method section.

AUTHORS:

With normalized counts, we mean the stellar counts normalized to the height of the Turn-on peak. The result is that in both panels the height of the TOn peak corresponds to 1. We clarified our normalization choice and changed the axis label to ‘Luminosity Function (normalized counts)’. However, we opted not to add a legend for the top two panels, as we believe it would increase the complexity of the plot.

The simulations were computed without field stars contamination. We added a sentence to explain how the simulation was derived.

-Figure A3:

This figure is wrongly labelled as Figure A2, while it should instead be A3. Also, it is not clear what the black dashed line refers to? It is not shown in the legend.

If it represents NGC 1818 observation ridge line, why this is not consistent with a ~40 Myr old population?

AUTHORS:

We thank the referee for pointing out the wrong labeling of the figures in the Method section. We noticed that the first Figure in the methods section is labeled A2 while it should be labeled as A1. We change the label of the first figure (pag. 20) to A1. Therefore, Figure A2 (pag.22 is now correctly labeled).

The black dashed line represents a Zero-Age Main-Sequence, while the colored lines represent isochrones of different ages as specified in the text. We added the label for the black dashed line

Figure 1. Analysis and Age Distribution computed for different value of the cluster radius, from 40 Arcsec to 52 Arcsec. The value of the adopted cluster radius is indicated in the top center.

Figure 2. *Top panel.* Cluster field and reference field, defined as the regions inside and outside the red and blue circles, respectively. When dealing with field contamination, we accounted for the different area of the cluster and reference field region. *Bottom panels.* Color-Magnitude Diagrams of the cluster field and reference field, respectively in the left and middle panel. The right panel shows the Luminosity Functions of cluster field stars (red line), i.e. all the stars enclosed by the red circle, reference field stars (blue line), and the subtracted Luminosity Function, obtained subtracting the blue LF from the red LF, shown in black. Shaded regions surrounding the black LF represent the uncertainties associated to each bin, determined with the bootstrap technique.

Referee: The reviewer's concerns and replies will be expressed in red below.

AUTHORS:

We thank the anonymous referees for their helpful comments and suggestions that have improved the quality of the manuscript. In the following, we answer each point individually, and we have modified the main text as discussed below (all changes in the main paper are marked with blue text).

REVIEWER COMMENTS

Reviewer #1 (Remarks to the Author):

2nd Review of Cordonì et al., The Turn-On of NGC1818 unveils the origin of Multiple Stellar Populations in Magellanic Cloud clusters; Nature Communications, re-submitted

I thank the authors for considering the issues I raised in my first report (Referee #1). I am mostly happy with the revised version of this paper, and I have no objections to it being published in Nature Communications.

Before recommending acceptance, however, I would like the authors to consider the following comments, provided as constructive criticism to help them further improve their paper.

1. There are still some instances of imprecise language. This is particularly obvious in the first, bold-face paragraph. That paragraph, although not formally speaking an abstract, will be read first by most people, so it should be self-explanatory and clear. In fact, the burden on the authors to make this as clear as possible is greatest for this paragraph. Here are some suggestions for further clarifications:

1a. (blue text) "... along the colour-magnitude diagrams ..." is imprecise. The multiple sequences are found alongside SPECIFIC FEATURES in CMDs. "Along CMDs" doesn't make sense here, I'm afraid.

1b. "... the Turn-Off of the [CMD] ..." This is also imprecise. It's not the turn-off of the CMD, but the turn-off of the main sequence in the CMD...

1c. "... of NGC1818 ..." This comes out of the blue. What is NGC 1818, why

is this an interesting object, etc. In the bold-face paragraph, you must make sure that the reader understand what the science is about, without having to go down a few pages to find out that the object is a young star cluster in the LMC (you should probably also include its age here).

AUTHORS:

We clarified the text following the reviewer's comments

Referee: Comments addressed.

2. On p. 2, where AGB stars are mentioned in association with a 50-150 Myr age spread, I would like to see references supporting those numbers; also, I was under the impression that AGB-induced age spreads are often suggested to encompass 300 Myr or even more, so why limit this to up to 150 Myr?

AUTHORS:

The upper limit of 150 Myr is imposed by the fact that low-mass AGB stars do not produce the right chemical composition for reproducing the 2G in mono-metallic GCs. We added references in the paper, while in the following we provide further details on why the AGB scenario excludes long star-formation durations.

Indeed, the classical AGB scenario suggests that only intermediate-mass AGB stars (i.e. 4 - 8 solar masses) are responsible for the formation of 2G stars. Following the detailed discussion in D'Ercole et al. (2010, MNRAS, 407, 854), the limiting age ranges from ~100 Myr (if second generations form from pure AGB ejecta) to ~150 Myr (in the case of dilution with pristine gas). Subsequent versions of the AGB scenario included super-AGB stars as additional polluters so that the lower limit to the age spread is about 50 Myr (e.g. Bastian & Lardo 2018 and references therein).

While the exact mass range is somehow model-dependent (e.g. Renzini et al. 2015, D'Ercole et al. 2010 for the caveats of their models), it is widely accepted that stars less massive than 3-4 solar masses can not be responsible for the formation of 2G stars.

Such constraint is due to the fact that the yields of low-mass AGB stars are strongly enhanced in the overall C+N+O abundance and in s-process elements, in contrast with what is observed in most GCs, where the

abundances of these elements are constant.

It should be noticed that in the case of the NGC1851, which is an anomalous GCs with internal variation in metallicity, s-process elements (Yong et al. 2008) and, possibly, in C+N+O (Yong et al. 2009, but see Tautvaišienė et al. 2022) low mass AGB stars have been invoked to explain the presence of the anomalous population. Similar arguments have been applied to the GCs M22 and omega Cen (e.g. Marino et al. 2012). In these cases, the time interval for the formation of the anomalous population would exceed 150 Myr, and supernovae, together with AGB, play a role in the SFH of the clusters.

In this work, we would prefer not to discuss the specific cases of GCs with heavy-element variations and refer to the classic AGB scenarios.

Referee: Ok, addressed.

3.p. 2, penultimate para (The eMSTOs have ..."): the authors argue that young MC clusters may be counterparts of old GCs. However, that is too simplistic. The peak of the GC mass function is at $2 \times 10^5 M_{\text{sun}}$, with very few clusters at masses of a few $\times 10^4 M_{\text{sun}}$, which is where these MC clusters come in. In addition, since these MC clusters are young, standard stellar evolution will make them lose at least 15% of their mass, if not more, over the course of their evolution to ages of 10-12 Gyr. Plus many will be dissolved dynamically given their low masses. As such, the chances that these MC clusters survive to become old GCs are really small. And even if they survive, their metallicities will be much higher than those of the current population of old GCs. These arguments taken together suggest to me that the young populous MC clusters are not really good counterparts to old Galactic GCs, even for comparison purposes...

AUTHORS:

We agree that young MC clusters and old Galactic GCs have different physical properties and would follow different dynamical evolution. Following the suggestions of the referee, we revised the text on comparison between young MC clusters and old Galactic GCs.

In particular, we deleted the sentence that MC could be the counterparts of GCs, which was misleading/incorrect, by quoting Conroy et al. (2011) and Keller et al. (2011), who suggested that a similar mechanism is responsible for the formation of multiple populations in eMSTO clusters and in old GCs

Moreover, even though young MC clusters and old Galactic GCs exhibit different physical properties, such as mass and metallicity, we believe that the connection between these two objects, and their multiple populations, is worth exploring. Specifically, we refer to the points discussed later in the present report, e.g. point 1 on pages 4 and 5.

Referee: For this point, see below the reply regarding this paragraph (para5 page 2), given by the second referee. In my opinion this paragraph has to be rephrased.

4.p. 3, top para: I agree that rotation on its own may not be sufficient to explain the extents of the eMSTOs. However, why do the authors categorically state that the remainder of the extents must come from age differences? What about metallicity differences or differences in the light-element abundances?

4a. The same comment applies to the first para in the section titled "The nature of the eMSTO" on p. 5.

AUTHORS:

Following Milone et al. (2016, e.g. their Fig. 9a-b), we exclude helium and iron variations as responsible for the eMSTO, because stellar populations with different Z or Y would not reproduce the observed eMSTO, MSs and RGBs. As an example, stellar populations with different helium abundances would result in split or broad RGB sequences, in contrast with the observations. Similarly, the merging of the MSs in young clusters is not consistent with significant differences in metals and helium.

We also exclude that the broadening/split of the eMSTO and bright MS stars are due to star-to-star variations in other light elements (such as C, N and O). Indeed, any variation in light elements, if present, would not provide a detectable change in the flux of these stars. Moreover, these elements would not affect the stellar fluxes in optical filters even in colder stars.

Because of these reasons, we assume that the only mechanism able to explain the residual spread in the Turn-Off region must be the age difference. We updated the main text clarifying this point.

Referee: Ok, this is addressed.

5. Throughout: I was a bit puzzled by the authors' use of the term "azure" for blue. While not technically incorrect, readers may not immediately know that meaning, so I recommend that they use "blue" instead.

AUTHORS:

We changed 'azure' with 'blue'.

Referee: Addressed.

6. In the caption of Fig. 2, I recommend that the authors add the step size between subsequent non-rotating isochrones in the top left panel.

AUTHORS:

We added the step size in the caption of Fig. 2

Referee: Addressed.

Reviewer #2 (Remarks to the Author):

Report for the paper "The Turn-On of NGC1818 unveils the origin of Multiple Stellar Populations in Magellanic Cloud clusters" by Cordoni et al.

The results reported in this paper are very interesting and extremely important in the context of understanding the phenomenon of the extended main sequence turn off (eMSTO) and split main sequence (MS) in star clusters younger than 2 Gyr.

I also believe that the analysis is robust and the technique used in the paper is well exploited for its purpose.

However, I have two major comments about the general discussion of the results and the context in which the paper was framed:

1) Throughout the paper, the authors claim that the reported results unveil the origin of multiple stellar populations, by connecting them with the multiple populations phenomenon observed in the ancient globular clusters (GCs). Multiple populations in old GCs manifest in the form of chemical abundance variations (He, Na, C, N etc), while in young star clusters no such chemical variations are observed to date (the youngest cluster where these are found in

the form of N spread is NGC 1783, ~1.7 Gyr old, see the recent paper from Cadelano et al. 2022, ApJ, 924L, 2C).

It is true and well established that young star clusters show extended main sequence turn offs and split main sequences in their CMDs that are definitely not “simple stellar populations”. Nevertheless, even if these phenomena and the multiple populations in old GCs might be related, a connection is not established, to date. Indeed, no spectroscopic measurements of chemical abundance variations along the eMSTO or split MS of young star clusters exist to prove this. Additionally, Milone et al. (2015, MNRAS, 450, 3750; 2016, MNRAS, 458, 4368) showed that stellar isochrones including chemical abundance spreads are not able to reproduce the observed bi-modal main sequences in young Magellanic Cloud clusters. A quote from their paper is “In contrast, isochrones with different helium abundance do not reproduce the observed CMD of NGC 1755. This fact provides a significant difference between the multiple sequences observed in the old Galactic GCs and in the young MC [Magellanic Cloud] clusters.”

In this context, I believe that the paper needs to be extensively re-written in order to connect the importance of its findings to a less generalised topic. The result of this paper is definitely a breakthrough into the debate between stellar rotation and age spreads to explain the phenomenon of the eMSTO and split MSs and it deserves publication. However, I found the connection with the multiple population phenomenon in the old GCs to be not relevant and, at the same time, confusing for a non-expert reader.

AUTHORS:

As suggested by the referee, we modified some text in the paper, including the abstract and the Introduction. We also clarify that our paper does not unveil the origin of multiple populations in GCs.

Nevertheless, we would prefer to keep some discussion on multiple populations in old GCs, and the conclusion that “multiple populations in GCs either coeval or formed with different mechanisms than the multiple populations of young clusters.”

Indeed, we believe that the notion of whether multiple populations in young and old clusters are similar phenomena or not is still an open and quite debated issue, and our work provides further constraints to the phenomenon.

In the following, we provide various arguments supporting our statement.

– **Chemical abundance variations** Chemical variations are one of the distinctive features of multiple populations in GCs and in several clusters older than about 1.5 Gyr.

The maximum internal variation in light elements in GCs correlates with cluster mass so that we expect that light element variations in low-mass clusters, if present, could be below the present detection threshold. We refer to Goudfrooij et al. (2014) for theoretical discussion.

This idea seems supported by the recent findings of multiple populations with different nitrogen abundance in the 1.5 Gyr old clusters NGC2173 (Kapse et al. 2022) and NGC1783 (Caledano et al. 2022). Here, the photometric detection of very-small nitrogen spread was allowed by the exquisite dataset, which had unprecedented quality in multiple-population studies of young clusters.

– **Age spreads** As mentioned now in the introduction, the small light-element variation, or even the lack thereof, could be consistent with a similar formation scenario in old and young clusters. If young clusters experienced prolonged star formation, it is reasonable to assume that clusters with chemical anomalies also host stellar populations of different ages. As an example, the recent discoveries of chemical anomalies in the eMSTO clusters NGC1783 and NGC2173 would fit in a scenario, where these clusters have experienced prolonged star formation and their N-rich stars are second-generation stars mostly from pristine material plus a small amount of ejecta from more massive 1G stars.

In general, scenarios for the formation of multiple populations in GCs, like the AGB scenario, suggest that 2G stars form from the mixing of pure ejecta and pristine material. The more pristine gas is accreted, i.e. more dilution, the more the chemical composition of 2G stars approaches that of 1G stars. The exact amount of pure ejecta and pristine material is poorly constrained and would depend on various quantities, including the mass of the polluter and the distribution of the pristine material.

Anyhow, in order to reproduce the abundance pattern observed in small-mass Galactic GC M4, where the internal sodium variation corresponds to ~ 0.4 dex and the mass fraction helium variation is 0.01 (e.g. Marino et al. 2008, Milone et al. 2018), most of the material from which 2G formed should be composed of pristine gas. By extending this scenario to the 2.5 Gyr old LMC cluster NGC1978, where the sodium variation between 1G and 2G stars is just 0.05 dex and there is virtually no helium variation (Saracino et al. 2020, Milone et al. 2020), we would conclude that only a tiny amount of AGB ejecta has been retained by the 2G star-forming region.

In this context, it would be reasonable to assume that a small amount of AGB ejecta has mixed with pristine material of the surrounding star-forming region, despite the smaller mass of this cluster.

By extending this argument to NGC1818, the ejecta could have been entirely, or widely, lost and the 2G would have formed from pristine material, such that the resulting abundance variations would be too small to be detected (we stress that with the available dataset, it would be not possible to detect in NGC1818 nitrogen variations that are as small as those observed in NGC1783 by Caledano et al. 2022). Hence, regardless of whether NGC1818 has retained AGB ejecta or not, it is crucial to constrain the duration of the star formation.

– **Stellar Mass** An additional reason why chemical anomalies are rarely detected in MC clusters younger than 2Gyr could be that most of the studies are based on RGB stars, which are more massive than about 1.5 solar mass. The AGB scenario by D’Ercole et al. (2008, 2010) assumes that multiple populations with different chemical composition only form below a certain mass threshold. Similarly, Bastian & Lardo (2018) suggested that chemical anomalies may originate among stars smaller than about 1.5 solar masses and are associated with the stellar structure.

The conclusion by Caledano et al. (2022) that chemical anomalies of NGC1783 are detected along the cluster MS (where stars have masses lower than 1.5 solar masses) but seem to be not present along the RGB is consistent with these predictions.

In this scenario, our result that 1 solar mass TOn stars of NGC1818 are nearly coeval would either challenge the AGB scenario or indicate that multiple populations in young clusters and in NGC1783 formed with different mechanisms.

– **Rotation** Milone et al. 2015, 2016 provided evidence of split MSs and eMSTO in young LMC clusters. Moreover, the 2016 paper on NGC1755 also shows that the split MS is not consistent with stellar populations with (large) variations in chemical composition but is well reproduced by isochrones with different rotation rates.

Nevertheless, a more accurate comparison between the observed and the simulated CMDs composed of stars with different rotation rates has revealed that the latter are not able to entirely reproduce the eMSTOs and that age difference, together with rotation, is needed to match the observation (see for e.g. Figure 10b in Milone et al. 2016, Figure 11 in Milone et al. 2017, Gossage et al. 2019 or the Goodfroom et al. 2017 paper ‘stellar rotation diminishes but does not eliminate age spread’).

Referee: I believe that the authors replied here that it is not demonstrated whether the same chemical anomalies are found in old and young star clusters. However, this is not a reply to my point raised above. My point is that, in the current paper, the authors are calculating the star formation history of the young star cluster NGC 1818, but they are not looking at whether multiple populations - in the form of chemical anomalies - are present in NGC 1818. This cannot be achieved with the current dataset, as expressed above, hence it is not demonstrated that NGC 1818 host multiple stellar populations. I still believe that interested readers will be confused in the way this paper is written, hence I suggest to remove any reference to the multiple stellar populations found in ancient GCs.

More specifically, I would suggest to:

- Modify the title into "...unveils the origin of the extended main sequence turn-off feature in young star clusters" or something similar.
- Start the paper with the paragraph on page 2 "Until a decade ago, young clusters...". I would indeed suggest not to start the paper with the multiple stellar populations phenomenon observed in old GCs, as the current paper does not infer anything on the presence of chemical variations in NGC 1818. The authors might keep the part about multiple stellar populations in ancient GCs as a separate discussion possibly, to express that a link between eMSTO and chemical variations might exist, although not established to date.
- Modify the paragraph that starts with "The eMSTOs have been immediately interpreted as..." as expressed in the specific point below in this report.
- Remove the last paragraph on page 5 "Since the early discovery, it has been suggested..."

The authors might also possibly keep a sentence about multiple populations by writing "If we assume that this cluster (NGC 1818) hosts multiple populations as chemical variations like in the ancient GCs, then these would have been formed very fast, almost concurrently", although I do not see the necessity for this.

Even if not demonstrated, NGC 1818 could indeed host chemical variations, but at the same time, it could also not show them. This needs to be corroborated with some other methods. In the current paper, the Turn-On is observed in optical filters, which are not sensitive to C, N, O variations. Additionally, there are many scenarios that do not envision a prolonged star formation within globular/star clusters for the formation of multiple populations (e.g., self-enrichment from massive and supermassive stars, e.g. de Mink et al. 2009, A&A, 507, L1, Denissenkov & Hartwick 2014, MNRAS, 437, L21, Gieles et al. 2018, MNRAS, 478, 2461, among many others). It has been also shown that there is no difference

in age between the multiple populations present in intermediate age star clusters (NGC 1978 and NGC 2121 in the LMC, the difference in age being less than $\sim 20\text{-}30$ Myr, Martocchia et al. 2018, MNRAS, 477, 4696, Saracino et al. 2020, 493, 6060S). Hence, a cluster that does not show a prolonged star formation could still be able to show chemical anomalies. It would be interesting to look for C/N/He spreads within such a cluster, but, as expressed already, it is not indeed possible with the current dataset.

I will accept the paper only if the authors refer their results to the phenomenon of the eMSTO/split MSs, without generalising to chemical variations in ancient GCs. I suggest to concentrate on the fact that the results of this paper are able to disprove for the first time the possibility that age spreads are the responsible feature for the phenomenon of the eMSTO and split MSs in young and intermediate age star clusters.

2) Despite being certainly a new and exciting result based on a novel technique, I found this paper to lack a few key references of works in which other authors attempted to find or disprove the presence of age spreads within young star clusters.

For example, Cabrera-Ziri et al. (2014, MNRAS, 441, 2754C), (2015, MNRAS, 448, 2224C), (2016, 457, 809C) found no evidence of extended star formation history in young star clusters in starburst galaxies, as well as Hollyhead et al. (2015, MNRAS, 449, 1106H).

A similar result to Cordoni et al. (2018, ApJ, 869, 139C) (cited in this paper), was found by Piatti & Bastian (2016, MNRAS, 463, 1632P) and Bastian et al. (2018, MNRAS, 480, 3739B) where they show that clusters that are not so massive (down to a few times $\sim 10^3$ Msun) also have eMSTOs. According to the age spread scenario for the origin of the eMSTO, these low mass clusters should not have been able to retain enough stellar ejecta to go through a prolonged period of star formation.

Furthermore, a key demonstration that extended star formation might be unlikely in clusters with an eMSTO was provided by the $\Delta(\text{Age})$ vs Age relation, from the work by Niederhofer et al. (2015, MNRAS, 453, 2070N). Such a relation showed that the older the cluster is, the larger the eMSTO width. Hence, this points towards a stellar evolutionary effect, most likely stellar rotation (which was nicely confirmed by Marino et al. 2018, ApJL, 863, L33, cited already in this paper). Another relevant paper is the one by Gossage et al. (2019, ApJ, 887, 199G), where the authors provide quantitative assessments of the eMSTO

morphology of several young clusters by modelling the effect of age spread, stellar rotation and both at the same time. They found that “a distribution of rotation rates appears to be the overall most physically motivated explanation for the eMSTO phenomenon”. The results from this paper are in agreement with what found by Gossage+19 and collaborators.

Hence, I suggest the authors to remodel the introduction as mentioned above in point 1), by also adding and discussing most of the references reported in point 2).

AUTHORS:

We modified the text as suggested and discussed all the results above.

As discussed in the paper, we agree that stellar rotation plays a major role in shaping the eMSTOs. Nevertheless, several papers conclude that rotation alone is not able to entirely reproduce the observed eMSTOs and some age spread is needed to match the observations (see Goudfrooij et al. 2017, Milone et al. 2017, Costa et al. 2018 and references therein). Even the detailed analysis by Gossage et al. 2019, ApJ 887 199 shows that while a distribution of rotation rates can reproduce the overall morphology of the eMSTO, a mixture of rotation rates and age spread is able to quantitatively match the observations.

It has been suggested that the inferred age spread is real (see papers by Goudfrooij and collaborators), or it is an artifact due to limitations in the available rotational models (e.g. Gossage et al. 2019) or it is due to phenomena associated with stellar breaking (e.g. D’Antona et al. 2018).

In any case, it is very challenging to disentangle among these possibilities by using the eMSTO alone. In contrast, the analysis of both the eMSTO and the MSTO introduced in this paper has the potential to address the question of whether eMSTO clusters host multiple stellar generations or not.

Referee: Ok for the references.

As reported in the introduction of the previous report and in the point above, I believe that the results presented in this paper are fundamental, i.e. the authors are presenting the first estimation (from resolved studies) of the age spread within a young star cluster. This is consistent with the lack of prolonged star formation, i.e. the presence of a single burst of star formation within the cluster, thus disproving the age spread scenario as the origin of the eMSTO and split MS in young and intermediate age star clusters. Hence, I do not question the importance of the paper. I question the connection with the “multiple stellar population” phenomenon observed in the old GCs. The authors write above as reply that this paper can “address the question of whether eMSTO clusters host multiple stellar

generations or not”. What do the authors mean here by “multiple generations”? Multiple bursts of star formation? If this is the case, I agree, the authors show in this paper that NGC 1818 do not host multiple stellar generations. If the authors refer to multiple generations as star-to-star chemical abundance variations - the so-called multiple populations - then I disagree with them. As reported in the introduction of this paper by the authors, it has been shown that the age difference between multiple populations in ancient GCs is estimated to be ~200 Myr or less, and this is the minimum constraint that we can obtain from ancient objects due to stellar evolution. The presence of chemical anomalies in NGC 1818 is not proved by the results of this paper, so we are actually not aware of the existence of multiple populations - as variations in chemistry - in this cluster. See the reply to the last point above, as well.

Below I report some additional comments throughout the text of the paper:

3A) first two paragraphs of the introduction:

As explained in the major points above, I suggest then to remove these paragraphs that connect old GCs and young star clusters and instead focus on the phenomena of eMSTO and split MS in young star clusters (<2 Gyr old). More generally, I also believe that a comparison with old GCs can be made throughout the text with a few sentences, as it is already done when stating that an estimation of age spreads in ancient GCs is hard because of stellar evolution (page 2, third paragraph).

3B) Introduction last paragraph: sentence “Hence, the debate is still entertained on whether the young clusters host multiple stellar generations or if the eMSTO and the multiple populations of old GCs are different phenomena”.

This is true but it is not relevant to the current study (see points above). The multiple stellar populations present in ancient GCs are star-to-star chemical abundance variations. We would need spectroscopic studies of stars along the eMSTO or split MSs to prove that also these phenomena show chemical abundance variations, at least.

AUTHORS:

See our reply to point 1 for discussion of the two points above.

Referee: These points were not addressed as well. See my reply to the last two points above, especially the former. The main point I raise is that with the current study the authors are not able to demonstrate that NGC 1818 hosts chemical variations. They estimate that NGC 1818 has a very fast star formation - although the cluster can still host multiple populations like the one present in the old GCs.

*-General comment about the nomenclature “multiple generations”:
Throughout the text, the authors often use “multiple generations” to refer to the multiple stellar populations, both for young and ancient clusters. This implies that a second (or third etc...) generation formed from the ejecta of a first generation of (massive) stars within the same cluster. Although being one of the possible scenario envisioned for the origin of the chemical abundance variations in old GCs, this model suffers from many unresolved issues, as well as other scenarios do. It is not established or proved that the multiple populations in old GCs are formed through this channel. Hence, I would prefer the authors to use more “neutral” terms such as “multiple stellar populations”, if they refer to ancient GCs.*

Secondly, “generation” or “multiple populations” should not be used to describe the populations present in NGC 1818 (such as the eMSTO and split MS) because it infers as well that the same type of populations are observed in eMSTO/split MS and in ancient GCs. I suggest to use “different populations” when referring to NGC 1818 or “the populations”, for instance.

AUTHORS:

We modified the wordings as suggested by the referee, changing ‘Multiple Generations’ with ‘Multiple Populations’ whenever needed.

Referee: Ok, see previous points for a rephrasing of the paper.

-page 2 paragraph named “NGC 1818. The ‘Rosetta stone’ to disentangle age and rotation.”

The authors state here that the “main properties of the multiple populations in young star clusters do not depend on cluster mass”. However, above, in page 2 paragraph 5, they stated that “the age spread inferred from the eMSTO correlated with cluster mass, in close analogy with what is observed in old GCs, where the complexity of multiple populations increases with the mass of the host cluster”.

Which main properties of young star clusters the authors referred to in the first sentence?

Also, the second sentence seems to contradict the first one.

Could you please elaborate more what you mean by these two sentences?

AUTHORS:

We clarified these points in the text, as suggested.

Referee: I see that this was explained, and the paragraph about the Rosetta Stone is fine. However, I am very concerned about what the authors write in the second

page, paragraph 5 (“The eMSTOs have been..”, same concern of Referee#1). The authors write that “young Magellanic Cloud clusters were considered “the missing link to understanding the evolution of old GCs with multiple generations”. They support this by writing that the inferred age spreads correlate with cluster mass in analogy to the multiple populations found in ancient GCs. Firstly, as explained in the points above, there is no link established between the phenomenon of eMSTO and the multiple populations (which is a spread in chemistry) found in old GCs. Secondly, this sentence contradicts the one reported in the next paragraph (The Rosetta stone), where the authors write that other features, such as the relative numbers of blue- and red- MS stars etc, do **not** correlate with cluster mass. I believe that this second sentence does not support what it is written above. To avoid confusion for the readers, my suggestion is to remove part of the paragraph and rephrase it in a way similar to this one:

“The eMSTOs have been immediately interpreted as the signature of prolonged star formation (e.g. 28, 9), as the age spread inferred from it can range from a few tens of Myr to ~500 Myr, (reference). Young Magellanic Cloud clusters were then immediately considered the missing link to understanding the evolution of old GCs with multiple POPULATIONS (e.g. 29, 30). “

And then continue with the next paragraph: *“However, the hypothesis ...[...]*

At first, the eMSTO phenomenon was considered as the same phenomenon of multiple populations present in old GCs, however no link is established between the two and most likely the eMSTO is due to stellar rotation. The results of the current work are crucial in disproving age spreads as responsible for the eMSTO.

-page 5, paragraph “The nature of the MSTO”

From “In addition to [...] disentangle the two scenarios”.

I believe that this sentence is very confusing and I strongly suggest to remove it, along with the references. The studies mentioned in the references (numbers 50, 51, 52) looked at multiple populations in intermediate age/young star clusters as chemical abundance variations and are most likely not linked to the phenomenon of the eMSTO, neither to stellar rotation.

AUTHORS:

We removed the sentence as suggested.

Referee: Ok.

-In Methods:

In Cignoni et al. (2010, ApJL, 712, L63), the authors list the sources of uncertainties arising when calculating the age spread from the Turn-On feature

of young star clusters. The majority of these uncertainties were considered in the current paper, however, has the differential reddening being checked for NGC 1818? I believe it should not significantly affect the result of the paper, but a few sentences about it would be useful in the Methods section.

AUTHORS:

We verified that NGC1818 is not affected by significant differential reddening, i.e. reddening variation, if present, is smaller than $E(B-V)=0.003$ mag, and is much smaller than photometric errors. We added a sentence in the Methods section.

Referee: Ok, addressed.

-Luminosity function paragraph in Methods:

The authors write about the procedure of subtracting the reference field stars to the cluster stars. How are they sure that they are not subtracting cluster stars and hence contaminating the results?

It would be nice to show here as reply to the report (probably there is no need in the paper) a map of the field of view of the observations with the selected regions for both cluster and reference stars, as well as the CMDs of the cluster and reference stars.

AUTHORS:

As discussed in the Methods section, to subtract the contribution of field stars we selected a region with radial distance from the cluster center larger than 75 arcsec, where can fairly assume that there are no cluster stars. As suggested, the cluster- and reference-field regions are shown in the top panel of Figure 2 of the present report, defined as the region inside and outside the red and blue circles, respectively.

Moreover, to investigate the effect of the field subtraction procedure, we repeated the analysis for different choices of the cluster field.

Figure 1 of this report illustrates the results for different cluster field radii, from 40 to 52 arcsec, and shows that the conclusions of the paper are not affected by this choice.

Referee: Ok, addressed.

-Figure 1:

On the x axis of the right panel the authors label "Normalised Counts". Does this represent the luminosity function? If not, how are these normalised counts

calculated? (the same applies to Figure 3). In order to clarify, I would suggest either to change the label to “Luminosity function(Counts)” or, if different, please explain in the text what “Normalised” means.

Also, does the CMD of the left panel of Figure 1 represent the field-stars cleaned CMD?

By comparing the histogram with the left panel of Figure 1 (binned CMD), it is possible to notice that around F814W mag of 21.8-22, the star counts are generally higher (black pixels) with respect to the Turn On peak (dark red pixels). Is this due to the fact that the width of the MS around the TOn peak is larger?

Generally, it would be useful to (i) add a colour bar of the star counts to the left panel, (ii) consistently plot the left and right panels, to show the field star subtracted CMD and counts.

AUTHORS:

As discussed below, we have modified Figure 1 of the manuscript including the bottom panels of Figure 2 of the present report. In the new version of the Figure, we did not normalize the histogram. Hence, the comment of the referee refers now to Figure 3 only (in the manuscript).

We clarify that stellar counts are normalised in such a way that the Turn-On peak corresponds to one. This choice is mostly for illustration purposes.

Referee: Ok, addressed.

The CMD shown in Figure 1 includes all the stars in the cluster field, without statistically subtracting field stars.

We emphasise here that we preferred not to statistically subtract field stars from the cluster-field CMD, because we accounted for field-star contamination when we calculated the LF of cluster stars. To do that, we subtracted the completeness-correct LF of field stars from the completeness-correct cluster-field LF (see Section Method for details). We clarified the text and the Figure as suggested. We confirm that the different pixel colour is due to the different MS width.

Referee: Why not showing the CMD with cluster stars that are field-subtracted? Have the authors checked how this would be different from the current CMD? I think that if this CMD also shows a peak in the Turn On, it could be an additional test to confirm the robustness of this analysis.

We also modified the CMD of Figure 1 to illustrate the contribution from field stars to the result, and, as suggested by the referee, we added the colourbar. In the rightmost panel, we show the completeness corrected and field subtracted

Luminosity Function (black lines), together with the reference field stars and cluster field stars LF, indicated by the blue and red lines, respectively.

-Figure 3:

I suggest to add a legend in the top two panels referring to the red and blue histograms. Additionally, the same comment of the point above is valid for the simulated histograms. What does “normalised” mean in this figure? Were the simulations calculated with a percentage of field stars or in this case field stars were not considered? In any case, I suggest to expand about it in the Method section.

AUTHORS:

With normalized counts, we mean the stellar counts normalized to the height of the Turn-on peak. The result is that in both panels the height of the TOn peak corresponds to 1. We clarified our normalization choice and changed the axis label to ‘Luminosity Function (normalized counts)’. However, we opted not to add a legend for the top two panels, as we believe it would increase the complexity of the plot.

The simulations were computed without field stars contamination. We added a sentence to explain how the simulation was derived.

Referee: Addressed.

-Figure A3:

This figure is wrongly labelled as Figure A2, while it should instead be A3. Also, it is not clear what the black dashed line refers to? It is not shown in the legend. If it represents NGC 1818 observation ridge line, why this is not consistent with a ~40 Myr old population?

AUTHORS:

We thank the referee for pointing out the wrong labeling of the figures in the Method section. We noticed that the first Figure in the methods section is labeled A2 while it should be labeled as A1. We change the label of the first figure (pag. 20) to A1. Therefore, Figure A2 (pag.22 is now correctly labeled).

The black dashed line represents a Zero-Age Main-Sequence, while the colored lines represent isochrones of different ages as specified in the text. We added the label for the black dashed line

Referee: Ok, addressed. In the caption of Figure A2, instead of black solid line, it has to be replaced with “black dashed line”.

Figure 1. Analysis and Age Distribution computed for different value of the cluster radius, from 40 Arcsec to 52 Arcsec. The value of the adopted cluster radius is indicated in the top center.

Referee: The reviewer's concerns and replies will be expressed in red below.

We thank the anonymous referee for his/her comments and suggestions. In the following, we reply to each referee's concern in blue text. For clarity, we leave our previous answers as black text. All changes in the manuscript are marked with blue text.

In particular, following the main suggestions from the referee, we removed the connection with old GCs along the entire paper.

We also replaced the bold-face introduction with a ~150 words non-referenced abstract, as indicated in the formatting instructions.

AUTHORS:

We thank the anonymous referees for their helpful comments and suggestions that have improved the quality of the manuscript. In the following, we answer each point individually, and we have modified the main text as discussed below (all changes in the main paper are marked with blue text).

REVIEWER COMMENTS

Reviewer #1 (Remarks to the Author):

2nd Review of Cordoni et al., The Turn-On of NGC1818 unveils the origin of Multiple Stellar Populations in Magellanic Cloud clusters; Nature Communications, re-submitted

I thank the authors for considering the issues I raised in my first report (Referee #1). I am mostly happy with the revised version of this paper, and I have no objections to it being published in Nature Communications.

Before recommending acceptance, however, I would like the authors to consider the following comments, provided as constructive criticism to help them further improve their paper.

1. There are still some instances of imprecise language. This is particularly obvious in the first, bold-face paragraph. That paragraph, although not formally speaking an abstract, will be read first by most people, so it should be self-explanatory and clear. In fact, the burden on the authors to make

this as clear as possible is greatest for this paragraph. Here are some suggestions for further clarifications:

1a. (blue text) "... along the colour-magnitude diagrams ..." is imprecise. The multiple sequences are found alongside SPECIFIC FEATURES in CMDs. "Along CMDs" doesn't make sense here, I'm afraid.

1b. "... the Turn-Off of the [CMD] ..." This is also imprecise. It's not the turn-off of the CMD, but the turn-off of the main sequence in the CMD...

1c. "... of NGC1818 ..." This comes out of the blue. What is NGC 1818, why is this an interesting object, etc. In the bold-face paragraph, you must make sure that the reader understand what the science is about, without having to go down a few pages to find out that the object is a young star cluster in the LMC (you should probably also include its age here).

AUTHORS:

We clarified the text following the reviewer's comments

Referee: Comments addressed.

2. On p. 2, where AGB stars are mentioned in association with a 50-150 Myr age spread, I would like to see references supporting those numbers; also, I was under the impression that AGB-induced age spreads are often suggested to encompass 300 Myr or even more, so why limit this to up to 150 Myr?

AUTHORS:

The upper limit of 150 Myr is imposed by the fact that low-mass AGB stars do not produce the right chemical composition for reproducing the 2G in mono metallic GCs. We added references in the paper, while in the following we provide further details on why the AGB scenario excludes long star-formation durations.

Indeed, the classical AGB scenario suggests that only intermediate-mass AGB stars (i.e. 4 - 8 solar masses) are responsible for the formation of 2G stars. Following the detailed discussion in D'Ercole et al. (2010, MNRAS, 407, 854), the limiting age ranges from ~100 Myr (if second generations form from pure AGB ejecta) to ~150 Myr (in the case of dilution with pristine gas). Subsequent versions of the AGB scenario included super-AGB stars as additional polluters

so that the lower limit to the age spread is about 50 Myr (e.g. Bastian & Lardo 2018 and references therein).

While the exact mass range is somehow model-dependent (e.g. Renzini et al. 2015, D'Ercole et al. 2010 for the caveats of their models), it is widely accepted that stars less massive than 3-4 solar masses can not be responsible for the formation of 2G stars.

Such constraint is due to the fact that the yields of low-mass AGB stars are strongly enhanced in the overall C+N+O abundance and in s-process elements, in contrast with what is observed in most GCs, where the abundances of these elements are constant.

It should be noticed that in the case of the NGC1851, which is an anomalous GCs with internal variation in metallicity, s-process elements (Yong et al. 2008) and, possibly, in C+N+O (Yong et al. 2009, but see Tautvaišienė et al. 2022) low mass AGB stars have been invoked to explain the presence of the anomalous population. Similar arguments have been applied to the GCs M22 and omega Cen (e.g. Marino et al. 2012). In these cases, the time interval for the formation of the anomalous population would exceed 150 Myr, and supernovae, together with AGB, play a role in the SFH of the clusters.

In this work, we would prefer not to discuss the specific cases of GCs with heavy-element variations and refer to the classic AGB scenarios.

Referee: Ok, addressed.

3.p. 2, penultimate para (The eMSTOs have ...''): the authors argue that young MC clusters may be counterparts of old GCs. However, that is too simplistic. The peak of the GC mass function is at $2 \times 10^5 M_{\text{sun}}$, with very few clusters at masses of a few $\times 10^4 M_{\text{sun}}$, which is where these MC clusters come in. In addition, since these MC clusters are young, standard stellar evolution will make them lose at least 15% of their mass, if not more, over the course of their evolution to ages of 10-12 Gyr. Plus many will be dissolved dynamically given their low masses. As such, the chances that these MC clusters survive to become old GCs are really small. And even if they survive, their metallicities will be much higher than those of the current population of old GCs. These arguments taken together suggest to me that the young populous MC clusters are not really good counterpartsto old Galactic GCs, even for comparison purposes...

AUTHORS:

We agree that young MC clusters and old Galactic GCs have different physical properties and would follow different dynamical evolution. Following the suggestions of the referee, we revised the text on comparison between young MC clusters and old Galactic GCs.

In particular, we deleted the sentence that MC could be the counterparts of GCs, which was misleading/incorrect, by quoting Conroy et al. (2011) and Keller et al. (2011), who suggested that a similar mechanism is responsible for the formation of multiple populations in eMSTO clusters and in old GCs. Moreover, even though young MC clusters and old Galactic GCs exhibit different physical properties, such as mass and metallicity, we believe that the connection between these two objects, and their multiple populations, is worth exploring. Specifically, we refer to the points discussed later in the present report, e.g. point 1 on pages 4 and 5.

Referee: For this point, see below the reply regarding this paragraph (para 5 page 2), given by the second referee. In my opinion this paragraph has to be rephrased.

AUTHORS: We rephrase the paragraph as suggested in the following report.

4.p. 3, top para: I agree that rotation on its own may not be sufficient to explain the extents of the eMSTOs. However, why do the authors categorically state that the remainder of the extents must come from age differences? What about metallicity differences or differences in the light element abundances?

4a. The same comment applies to the first para in the section titled "The nature of the eMSTO" on p. 5.

AUTHORS:

Following Milone et al. (2016, e.g. their Fig. 9a-b), we exclude helium and iron variations as responsible for the eMSTO, because stellar populations with different Z or Y would not reproduce the observed eMSTO, MSs and RGBs. As an example, stellar populations with different helium abundances would result in split or broad RGB sequences, in contrast with the observations. Similarly, the merging of the MSs in young clusters is not consistent with significant differences in metals and helium.

We also exclude that the broadening/split of the eMSTO and bright MS stars are due to star-to-star variations in other light elements (such as C, N and O). Indeed, any variation in light elements, if present, would not provide a detectable change in the flux of these stars. Moreover, these elements would not affect the stellar fluxes in optical filters even in colder stars.

Because of these reasons, we assume that the only mechanism able to explain the residual spread in the Turn-Off region must be the age difference. We updated the main text clarifying this point.

Referee: Ok, this is addressed.

5.Throughout: I was a bit puzzled by the authors' use of the term "azure" for blue. While not technically incorrect, readers may not immediately know that meaning, so I recommend that they use "blue" instead.

AUTHORS:

We changed ‘azure’ with ‘blue’.

Referee: Addressed.

6.In the caption of Fig. 2, I recommend that the authors add the step size between subsequent non-rotating isochrones in the top left panel.

AUTHORS:

We added the step size in the caption of Fig. 2

Referee: Addressed.

Reviewer #2 (Remarks to the Author):

Report for the paper “The Turn-On of NGC1818 unveils the origin of Multiple Stellar Populations in Magellanic Cloud clusters” by Cordoni et al.

The results reported in this paper are very interesting and extremely important in the context of understanding the phenomenon of the extended main sequence turn off (eMSTO) and split main sequence (MS) in star clusters younger than 2 Gyr.

I also believe that the analysis is robust and the technique used in the paper is well exploited for its purpose.

However, I have two major comments about the general discussion of the results and the context in which the paper was framed:

1) Throughout the paper, the authors claim that the reported results unveil the origin of multiple stellar populations, by connecting them with the multiple populations phenomenon observed in the ancient globular clusters (GCs). Multiple populations in old GCs manifest in the form of chemical abundance variations (He, Na, C, N etc), while in young star clusters no such chemical variations are observed to date (the youngest cluster where these are found in the form of N spread is NGC 1783, ~1.7 Gyr old, see the recent paper from Cadelano et al. 2022, ApJ, 924L, 2C).

It is true and well established that young star clusters show extended main sequence turn offs and split main sequences in their CMDs that are definitely not “simple stellar populations”. Nevertheless, even if these phenomena and the multiple populations in old GCs might be related, a connection is not established, to date. Indeed, no spectroscopic measurements of chemical abundance variations along the eMSTO or split MS of young star clusters exist to prove this. Additionally, Milone et al. (2015, MNRAS, 450, 3750; 2016, MNRAS, 458, 4368) showed that stellar isochrones including chemical abundance spreads are not able to reproduce the observed bi-modal main sequences in young Magellanic Cloud clusters. A quote from their paper is “In contrast, isochrones with different helium abundance do not reproduce the observed CMD of NGC 1755. This fact provides a significant difference between the multiple sequences observed in the old Galactic GCs and in the young MC [Magellanic Cloud] clusters.”

In this context, I believe that the paper needs to be extensively re-written in order to connect the importance of its findings to a less generalised topic. The result of this paper is definitely a breakthrough into the debate between stellar rotation and age spreads to explain the phenomenon of the eMSTO and split MSs and it deserves publication. However, I found the connection with the multiple population phenomenon in the old GCs to be not relevant and, at the same time, confusing for a non-expert reader.

AUTHORS:

As suggested by the referee, we modified some text in the paper, including the abstract and the Introduction. We also clarify that our paper does not unveil the origin of multiple populations in GCs.

Nevertheless, we would prefer to keep some discussion on multiple populations in old GCs, and the conclusion that “multiple populations in GCs either coeval or formed with different mechanisms than the multiple populations of young

clusters.”

Indeed, we believe that the notion of whether multiple populations in young and old clusters are similar phenomena or not is still an open and quite debated issue, and our work provides further constraints to the phenomenon.

In the following, we provide various arguments supporting our statement.

– **Chemical abundance variations** Chemical variations are one of the distinctive features of multiple populations in GCs and in several clusters older than about 1.5 Gyr.

The maximum internal variation in light elements in GCs correlates with cluster mass so that we expect that light element variations in low-mass clusters, if present, could be below the present detection threshold. We refer to Goudfrooij et al. (2014) for theoretical discussion.

This idea seems supported by the recent findings of multiple populations with different nitrogen abundance in the 1.5 Gyr old clusters NGC2173 (Kapse et al. 2022) and NGC1783 (Caledano et al. 2022). Here, the photometric detection of very-small nitrogen spread was allowed by the exquisite dataset, which had unprecedented quality in multiple-population studies of young clusters.

– **Age spreads** As mentioned now in the introduction, the small light-element variation, or even the lack thereof, could be consistent with a similar formation scenario in old and young clusters. If young clusters experienced prolonged star formation, it is reasonable to assume that clusters with chemical anomalies also host stellar populations of different ages. As an example, the recent discoveries of chemical anomalies in the eMSTO clusters NGC1783 and NGC2173 would fit in a scenario, where these clusters have experienced prolonged star formation and their N-rich stars are second-generation stars mostly from pristine material plus a small amount of ejecta from more massive 1G stars.

In general, scenarios for the formation of multiple populations in GCs, like the AGB scenario, suggest that 2G stars form from the mixing of pure ejecta and pristine material. The more pristine gas is accreted, i.e. more dilution, the more the chemical composition of 2G stars approaches that of 1G stars. The exact amount of pure ejecta and pristine material is poorly constrained and would depend on various quantities, including the mass of the polluter and the distribution of the pristine material.

Anyhow, in order to reproduce the abundance pattern observed in small-mass Galactic GC M4, where the internal sodium variation corresponds to ~ 0.4 dex and the mass fraction helium variation is 0.01 (e.g. Marino et al. 2008, Milone et al. 2018), most of the material from which 2G formed should be composed of pristine gas. By extending this scenario to the 2.5 Gyr old LMC cluster NGC1978, where the sodium variation between 1G and 2G stars is just 0.05 dex and there is virtually no helium variation (Saracino et al. 2020, Milone et al.

2020), we would conclude that only a tiny amount of AGB ejecta has been retained by the 2G star-forming region.

In this context, it would be reasonable to assume that a small amount of AGB ejecta has mixed with pristine material of the surrounding star-forming region, despite the smaller mass of this cluster.

By extending this argument to NGC1818, the ejecta could have been entirely, or widely, lost and the 2G would have formed from pristine material, such that the resulting abundance variations would be too small to be detected (we stress that with the available dataset, it would be not possible to detect in NGC1818 nitrogen variations that are as small as those observed in NGC1783 by Caledano et al. 2022). Hence, regardless of whether NGC1818 has retained AGB ejecta or not, it is crucial to constrain the duration of the star formation.

– **Stellar Mass** An additional reason why chemical anomalies are rarely detected in MC clusters younger than 2Gyr could be that most of the studies are based on RGB stars, which are more massive than about 1.5 solar mass. The AGB scenario by D’Ercole et al. (2008, 2010) assumes that multiple populations with different chemical composition only form below a certain mass threshold. Similarly, Bastian & Lardo (2018) suggested that chemical anomalies may originate among stars smaller than about 1.5 solar masses and are associated with the stellar structure.

The conclusion by Caledano et al. (2022) that chemical anomalies of NGC1783 are detected along the cluster MS (where stars have masses lower than 1.5 solar masses) but seem to be not present along the RGB is consistent with these predictions.

In this scenario, our result that 1 solar mass TOn stars of NGC1818 are nearly coeval would either challenge the AGB scenario or indicate that multiple populations in young clusters and in NGC1783 formed with different mechanisms.

– **Rotation** Milone et al. 2015, 2016 provided evidence of split MSs and eMSTO in young LMC clusters. Moreover, the 2016 paper on NGC1755 also shows that the split MS is not consistent with stellar populations with (large) variations in chemical composition but is well reproduced by isochrones with different rotation rates.

Nevertheless, a more accurate comparison between the observed and the simulated CMDs composed of stars with different rotation rates has revealed that the latter are not able to entirely reproduce the eMSTOs and that age difference, together with rotation, is needed to match the observation (see for e.g. Figure 10b in Milone et al. 2016, Figure 11 in Milone et al. 2017, Gossage et al. 2019 or the Goodfroom et al. 2017 paper ‘stellar rotation diminishes but does not eliminate age spread’).

Referee: I believe that the authors replied here that it is not demonstrated whether the same chemical anomalies are found in old and young star clusters. However, this is not a reply to my point raised above. My point is that, in the current paper, the authors are calculating the star formation history of the young star cluster NGC 1818, but they are not looking at whether multiple populations - in the form of chemical anomalies - are present in NGC 1818. This cannot be achieved with the current dataset, as expressed above, hence it is not demonstrated that NGC 1818 host multiple stellar populations. I still believe that interested readers will be confused in the way this paper is written, hence I suggest to remove any reference to the multiple stellar populations found in ancient GCs.

More specifically, I would suggest to:

- Modify the title into "...unveils the origin of the extended main sequence turn off feature in young star clusters" or something similar.

AUTHORS: We modified the title as suggested

- Start the paper with the paragraph on page 2 "Until a decade ago, young clusters...". I would indeed suggest not to start the paper with the multiple stellar populations phenomenon observed in old GCs, as the current paper does not infer anything on the presence of chemical variations in NGC 1818. The authors might keep the part about multiple stellar populations in ancient GCs as a separate discussion possibly, to express that a link between eMSTO and chemical variations might exist, although not established to date.

AUTHORS: We removed the first paragraphs of the Introduction.

- Modify the paragraph that starts with "The eMSTOs have been immediately interpreted as..." as expressed in the specific point below in this report.

AUTHORS: We modified the paragraph as suggested

- Remove the last paragraph on page 5 "Since the early discovery, it has been suggested..."

AUTHORS: We removed the paragraph as suggested

The authors might also possibly keep a sentence about multiple populations by writing "If we assume that this cluster (NGC 1818) hosts multiple populations as chemical variations like in the ancient GCs, then these would have been formed very fast, almost concurrently", although I do not see the necessity for this.

Even if not demonstrated, NGC 1818 could indeed host chemical variations, but at the same time, it could also not show them. This needs to be corroborated with some other methods. In the current paper, the Turn-On is observed in optical filters, which are not sensitive to C, N, O variations. Additionally, there are many

scenarios that do not envision a prolonged star formation within globular/star clusters for the formation of multiple populations (e.g., self-enrichment from massive and supermassive stars, e.g. de Mink et al. 2009, A&A, 507, L1, Denissenkov & Hartwick 2014, MNRAS, 437, L21, Gieles et al. 2018, MNRAS, 478, 2461, among many others). It has been also shown that there is no difference in age between the multiple populations present in intermediate age star clusters (NGC 1978 and NGC 2121 in the LMC, the difference in age being less than ~ 20 -30 Myr, Martocchia et al. 2018, MNRAS, 477, 4696, Saracino et al. 2020, 493, 6060S). Hence, a cluster that does not show a prolonged star formation could still be able to show chemical anomalies. It would be interesting to look for C/N/He spreads within such a cluster, but, as expressed already, it is not indeed possible with the current dataset.

I will accept the paper only if the authors refer their results to the phenomenon of the eMSTO/split MSs, without generalising to chemical variations in ancient GCs. I suggest to concentrate on the fact that the results of this paper are able to disprove for the first time the possibility that age spreads are the responsible feature for the phenomenon of the eMSTO and split MSs in young and intermediate age star clusters.

AUTHORS: Following the referee's suggestions, we rescoped the manuscript to focus on the multipopulations phenomenon in Magellanic Clouds clusters, i.e. eMSTOs and split-MSs

2) Despite being certainly a new and exciting result based on a novel technique, I found this paper to lack a few key references of works in which other authors attempted to find or disprove the presence of age spreads within young star clusters.

For example, Cabrera-Ziri et al. (2014, MNRAS, 441, 2754C), (2015, MNRAS, 448, 2224C), (2016, 457, 809C) found no evidence of extended star formation history in young star clusters in starburst galaxies, as well as Hollyhead et al. (2015, MNRAS, 449, 1106H).

A similar result to Cordoni et al. (2018, ApJ, 869, 139C) (cited in this paper), was found by Piatti & Bastian (2016, MNRAS, 463, 1632P) and Bastian et al. (2018, MNRAS, 480, 3739B) where they show that clusters that are not so massive (down to a few times $\sim 10^3$ Msun) also have eMSTOs. According to the age spread scenario for the origin of the eMSTO, these low mass clusters should not have been able to retain enough stellar ejecta to go through a prolonged period of star formation.

Furthermore, a key demonstration that extended star formation might be unlikely in clusters with an eMSTO was provided by the $\Delta(\text{Age})$ vs Age relation, from

the work by Niederhofer et al. (2015, MNRAS, 453, 2070N). Such a relation showed that the older the cluster is, the larger the eMSTO width. Hence, this points towards a stellar evolutionary effect, most likely stellar rotation (which was nicely confirmed by Marino et al. 2018, ApJL, 863, L33, cited already in this paper). Another relevant paper is the one by Gossage et al. (2019, ApJ, 887, 199G), where the authors provide quantitative assessments of the eMSTO morphology of several young clusters by modelling the effect of age spread, stellar rotation and both at the same time. They found that “a distribution of rotation rates appears to be the overall most physically motivated explanation for the eMSTO phenomenon”. The results from this paper are in agreement with what found by Gossage+19 and collaborators.

Hence, I suggest the authors to remodel the introduction as mentioned above in point 1), by also adding and discussing most of the references reported in point 2).

AUTHORS:

We modified the text as suggested and discussed all the results above.

As discussed in the paper, we agree that stellar rotation plays a major role in shaping the eMSTOs. Nevertheless, several papers conclude that rotation alone is not able to entirely reproduce the observed eMSTOs and some age spread is needed to match the observations (see Goudfrooij et al. 2017, Milone et al. 2017, Costa et al. 2018 and references therein). Even the detailed analysis by Gossage et al. 2019, ApJ 887 199 shows that while a distribution of rotation rates can reproduce the overall morphology of the eMSTO, a mixture of rotation rates and age spread is able to quantitatively match the observations.

It has been suggested that the inferred age spread is real (see papers by Goudfrooij and collaborators), or it is an artifact due to limitations in the available rotational models (e.g. Gossage et al. 2019) or it is due to phenomena associated with stellar breaking (e.g. D’Antona et al. 2018).

In any case, it is very challenging to disentangle among these possibilities by using the eMSTO alone. In contrast, the analysis of both the eMSTO and the MSTO introduced in this paper has the potential to address the question of whether eMSTO clusters host multiple stellar generations or not.

Referee: Ok for the references.

As reported in the introduction of the previous report and in the point above, I believe that the results presented in this paper are fundamental, i.e. the authors are presenting the first estimation (from resolved studies) of the age spread within a young star cluster. This is consistent with the lack of prolonged star formation, i.e. the presence of a single burst of star formation within the cluster, thus disproving

the age spread scenario as the origin of the eMSTO and split MS in young and intermediate age star clusters. Hence, I do not question the importance of the paper. I question the connection with the “multiple stellar population” phenomenon observed in the old GCs. The authors write above as reply that this paper can “address the question of whether eMSTO clusters host multiple stellar generations or not”. What do the authors mean here by “multiple generations”? Multiple bursts of star formation? If this is the case, I agree, the authors show in this paper that NGC 1818 do not host multiple stellar generations. If the authors refer to multiple generations as star-to-star chemical abundance variations - the so-called multiple populations - then I disagree with them. As reported in the introduction of this paper by the authors, it has been shown that the age difference between multiple populations in ancient GCs is estimated to be ~200 Myr or less, and this is the minimum constraint that we can obtain from ancient objects due to stellar evolution. The presence of chemical anomalies in NGC 1818 is not proved by the results of this paper, so we are actually not aware of the existence of multiple populations - as variations in chemistry - in this cluster. See the reply to the last point above, as well.

AUTHORS: To account for the referee’s comments, we removed all discussion about the possible connection with multiple populations in old GCs.

Below I report some additional comments throughout the text of the paper:

3A) first two paragraphs of the introduction:

As explained in the major points above, I suggest then to remove these paragraphs that connect old GCs and young star clusters and instead focus on the phenomena of eMSTO and split MS in young star clusters (<2 Gyr old). More generally, I also believe that a comparison with old GCs can be made throughout the text with a few sentences, as it is already done when stating that an estimation of age spreads in ancient GCs is hard because of stellar evolution (page 2, third paragraph).

3B) Introduction last paragraph: sentence “Hence, the debate is still entertained on whether the young clusters host multiple stellar generations or if the eMSTO and the multiple populations of old GCs are different phenomena”. This is true but it is not relevant to the current study (see points above). The multiple stellar populations present in ancient GCs are star-to-star chemical abundance variations. We would need spectroscopic studies of stars along the eMSTO or split MSs to prove that also these phenomena show chemical abundance variations, at least.

AUTHORS:

See our reply to point 1 for discussion of the two points above.

Referee: These points were not addressed as well. See my reply to the last two points above, especially the former. The main point I raise is that with the current study the authors are not able to demonstrate that NGC 1818 hosts chemical variations. They estimate that NGC 1818 has a very fast star formation - although the cluster can still host multiple populations like the one present in the old GCs.

AUTHORS: As suggested we removed references to multiple stellar populations in old GCs.

-General comment about the nomenclature “multiple generations”: Throughout the text, the authors often use “multiple generations” to refer to the multiple stellar populations, both for young and ancient clusters. This implies that a second (or third etc...) generation formed from the ejecta of a first generation of (massive) stars within the same cluster. Although being one of the possible scenarios envisioned for the origin of the chemical abundance variations in old GCs, this model suffers from many unresolved issues, as well as other scenarios do. It is not established or proved that the multiple populations in old GCs are formed through this channel. Hence, I would prefer the authors to use more “neutral” terms such as “multiple stellar populations”, if they refer to ancient GCs.

Secondly, “generation” or “multiple populations” should not be used to describe the populations present in NGC 1818 (such as the eMSTO and split MS) because it infers as well that the same type of populations are observed in eMSTO/split MS and in ancient GCs. I suggest to use “different populations” when referring to NGC 1818 or “the populations”, for instance.

AUTHORS:

We modified the wordings as suggested by the referee, changing ‘Multiple Generations’ with ‘Multiple Populations’ whenever needed.

Referee: Ok, see previous points for a rephrasing of the paper.

-page 2 paragraph named “NGC 1818. The ‘Rosetta stone’ to disentangle age and rotation.”

The authors state here that the “main properties of the multiple populations in young star clusters do not depend on cluster mass”. However, above, in page 2 paragraph 5, they stated that “the age spread inferred from the eMSTO correlated with cluster mass, in close analogy with what is observed in old GCs, where the complexity of multiple populations increases with the mass of the host cluster”.

Which main properties of young star clusters the authors referred to in the first

sentence?

Also, the second sentence seems to contradict the first one.

Could you please elaborate more what you mean by these two sentences?

AUTHORS:

We clarified these points in the text, as suggested.

Referee: I see that this was explained, and the paragraph about the Rosetta Stone is fine. However, I am very concerned about what the authors write in the second page, paragraph 5 (“The eMSTOs have been..”, same concern of Referee#1). The authors write that “young Magellanic Cloud clusters were considered “the missing link to understanding the evolution of old GCs with multiple generations”. They support this by writing that the inferred age spreads correlate with cluster mass in analogy to the multiple populations found in ancient GCs. Firstly, as explained in the points above, there is no link established between the phenomenon of eMSTO and the multiple populations (which is a spread in chemistry) found in old GCs. Secondly, this sentence contradicts the one reported in the next paragraph (The Rosetta stone), where the authors write that other features, such as the relative numbers of blue- and red- MS stars etc, do *not* correlate with cluster mass. I believe that this second sentence does not support what it is written above. To avoid confusion for the readers, my suggestion is to remove part of the paragraph and rephrase it in a way similar to this one:

“The eMSTOs have been immediately interpreted as the signature of prolonged star formation (e.g. 28, 9), as the age spread inferred from it can range from a few tens of Myr to ~500 Myr, (reference). Young Magellanic Cloud clusters were then immediately considered the missing link to understanding the evolution of old GCs with multiple POPULATIONS (e.g. 29, 30). “

AUTHORS: We modified the paragraph as suggested.

And then continue with the next paragraph: *“However, the hypothesis ...[...]*

At first, the eMSTO phenomenon was considered as the same phenomenon of multiple populations present in old GCs, however no link is established between the two and most likely the eMSTO is due to stellar rotation. The results of the current work are crucial in disproving age spreads as responsible for the eMSTO.

-page 5, paragraph “The nature of the MSTO”

From “In addition to [...] disentangle the two scenarios”.

I believe that this sentence is very confusing and I strongly suggest to remove it, along with the references. The studies mentioned in the references (numbers 50, 51, 52) looked at multiple populations in intermediate age/young star clusters as

chemical abundance variations and are most likely not linked to the phenomenon of the eMSTO, neither to stellar rotation.

AUTHORS:

We removed the sentence as suggested.

Referee: Ok.

-In Methods:

In Cignoni et al. (2010, ApJL, 712, L63), the authors list the sources of uncertainties arising when calculating the age spread from the Turn-On feature of young star clusters. The majority of these uncertainties were considered in the current paper, however, has the differential reddening being checked for NGC 1818? I believe it should not significantly affect the result of the paper, but a few sentences about it would be useful in the Methods section.

AUTHORS:

We verified that NGC1818 is not affected by significant differential reddening, i.e. reddening variation, if present, is smaller than $E(B-V)=0.003$ mag, and is much smaller than photometric errors. We added a sentence in the Methods section.

Referee: Ok, addressed.

-Luminosity function paragraph in Methods:

The authors write about the procedure of subtracting the reference field stars to the cluster stars. How are they sure that they are not subtracting cluster stars and hence contaminating the results?

It would be nice to show here as reply to the report (probably there is no need in the paper) a map of the field of view of the observations with the selected regions for both cluster and reference stars, as well as the CMDs of the cluster and reference stars.

AUTHORS:

As discussed in the Methods section, to subtract the contribution of field stars we selected a region with radial distance from the cluster center larger than 75 arcsec, where can fairly assume that there are no cluster stars. As suggested, the cluster and reference-field regions are shown in the top panel of Figure 2 of the present report, defined as the region inside and outside the red and blue circles, respectively.

Moreover, to investigate the effect of the field subtraction procedure, we repeated

the analysis for different choices of the cluster field.

Figure 1 of this report illustrates the results for different cluster field radii, from 40 to 52 arcsec, and shows that the conclusions of the paper are not affected by this choice.

Referee: Ok, addressed.

-Figure 1:

On the x axis of the right panel the authors label “Normalised Counts”. Does this represent the luminosity function? If not, how are these normalised counts calculated? (the same applies to Figure 3). In order to clarify, I would suggest either to change the label to “Luminosity function(Counts)” or, if different, please explain in the text what “Normalised” means.

Also, does the CMD of the left panel of Figure 1 represent the field-stars cleaned CMD?

By comparing the histogram with the left panel of Figure 1 (binned CMD), it is possible to notice that around F814W mag of 21.8-22, the star counts are generally higher (black pixels) with respect to the Turn On peak (dark red pixels). Is this due to the fact that the width of the MS around the TOn peak is larger?

Generally, it would be useful to (i) add a colour bar of the star counts to the left panel, (ii) consistently plot the left and right panels, to show the field star subtracted CMD and counts.

AUTHORS:

As discussed below, we have modified Figure 1 of the manuscript including the bottom panels of Figure 2 of the present report. In the new version of the Figure, we did not normalize the histogram. Hence, the comment of the referee refers now to Figure 3 only (in the manuscript).

We clarify that stellar counts are normalised in such a way that the Turn-On peak corresponds to one. This choice is mostly for illustration purposes.

Referee: Ok, addressed.

The CMD shown in Figure 1 includes all the stars in the cluster field, without statistically subtracting field stars.

We emphasise here that we preferred not to statistically subtract field stars from the cluster-field CMD, because we accounted for field-star contamination when we calculated the LF of cluster stars. To do that, we subtracted the completeness correct LF of field stars from the completeness-correct cluster-field LF (see Section Method for details). We clarified the text and the Figure as suggested. We confirm that the different pixel colour is due to the different MS width.

Referee: Why not showing the CMD with cluster stars that are field-subtracted? Have the authors checked how this would be different from the current CMD? I think that if this CMD also shows a peak in the Turn On, it could be an additional test to confirm the robustness of this analysis.

AUTHORS: We show in the present report the CMD of the field-subtracted cluster region. Specifically, we name “Method 1” the subtraction procedure described in the Methods section, i.e. performed on the Luminosity Function, and “Method 2” the procedure suggested by the referee, i.e. subtraction performed on the CMD before computing the Luminosity Function. In both cases, we accounted for incompleteness. The results of the procedures are shown in the following Figure 1. As expected, Method 1 and Method 2 return nearly identical results, corroborating our analysis and results.

We briefly summarized results from this test in the Methods section of the manuscript.

Figure 1. *Left panel.* Field subtracted Color-Magnitude Diagrams of the cluster field. *Middle panel.* Subtracted stars from the cluster field. *Right panel.* Field subtracted Luminosity Functions derived with Method 1 (black line), i.e. field subtraction performed on the LF, as discussed in the Methods section of the manuscript, and with Method 2 (light blue line), i.e. subtraction performed in the CMD as suggested by the referee. Shaded regions surrounding the black LF represent the uncertainties associated to each bin, determined with the bootstrap technique.

We also modified the CMD of Figure 1 to illustrate the contribution from field stars to the result, and, as suggested by the referee, we added the colourbar. In the rightmost panel, we show the completeness corrected and field subtracted Luminosity Function (black lines), together with the reference field stars and

cluster field stars LF, indicated by the blue and red lines, respectively.

-Figure 3:

I suggest to add a legend in the top two panels referring to the red and blue histograms. Additionally, the same comment of the point above is valid for the simulated histograms. What does “normalised” mean in this figure? Were the simulations calculated with a percentage of field stars or in this case field stars were not considered? In any case, I suggest to expand about it in the Method section.

AUTHORS:

With normalized counts, we mean the stellar counts normalized to the height of the Turn-on peak. The result is that in both panels the height of the TOn peak corresponds to 1. We clarified our normalization choice and changed the axis label to ‘Luminosity Function (normalized counts)’. However, we opted not to add a legend for the top two panels, as we believe it would increase the complexity of the plot.

The simulations were computed without field stars contamination. We added a sentence to explain how the simulation was derived.

Referee: Addressed.

-Figure A3:

This figure is wrongly labelled as Figure A2, while it should instead be A3. Also, it is not clear what the black dashed line refers to? It is not shown in the legend. If it represents NGC 1818 observation ridge line, why this is not consistent with a ~40 Myr old population?

AUTHORS:

We thank the referee for pointing out the wrong labeling of the figures in the Method section. We noticed that the first Figure in the methods section is labeled A2 while it should be labeled as A1. We change the label of the first figure (pag. 20) to A1. Therefore, Figure A2 (pag.22 is now correctly labeled).

The black dashed line represents a Zero-Age Main-Sequence, while the colored lines represent isochrones of different ages as specified in the text. We added the label for the black dashed line

Referee: Ok, addressed. In the caption of Figure A2, instead of black solid line, it has to be replaced with “black dashed line”.

AUTHORS: We corrected the caption

Figure 1. Analysis and Age Distribution computed for different value of the cluster radius, from 40 Arcsec to 52 Arcsec. The value of the adopted cluster radius is indicated in the top center.

REVIEWERS' COMMENTS

Reviewer #2 (Remarks to the Author):

I really appreciated the new version of the paper which is much more solid and clear, in my opinion.

I am ready to accept the manuscript, however I would only like the authors to take these final considerations into account, reported below. More specifically, for the manuscript to be as clearest as possible, I would like the authors not to refer to this phenomenon as “multiple stellar populations”, as it is known that the multiple populations phenomenon is the one associated with the old GCs (the chemistry variations). Find below all the changes and some suggestions for replacement:

Abstract

Line 9 - remove “multiple stellar populations” -> The origin of star clusters , or young star clusters

Line 19 - remove “multiple populations” -> “stellar populations in young clusters”

Introduction

Line 23 - remove “multiple stellar populations” -> the discovery of multiple sequences etc

Line 37 - It could be useful to explain with a few words what are these multiple populations: “...with multiple stellar populations, i.e. star-to-star light-element abundance variations (10,11)”.

Line 66-67 - remove multiple stellar populations & Line 68- remove multipopulation phenomenon

You can possibly change this sentence to: “NGC 1818 represents the perfect target to unveil the origin of the eMSTO as it is very young and shows the typical features found in young Magellanic Clouds clusters' CMDs”.

Line 70 remove “multiple populations “ ->“main properties of young clusters”

Line 73 remove “multiple populations”

Line 103 - remove “multiple populations” - replace with eMSTO - split MSs... or the multiple sequences in the CMDs of young clusters.

Line 111 - remove “multiple populations” - ... has never been used to estimate the star formation history of young and massive Magellanic Cloud clusters stellar populations.

Line 165 - remove “multiple”

Line 171 - remove “multipopulations” - For example, replace with “star formation history” or the origin of the eMSTO

Also, the text has some problems of visualisation. The equations are not visible anymore.

Fourth Report
REVIEWERS' COMMENTS

Reviewer #2 (Remarks to the Author):

I really appreciated the new version of the paper which is much more solid and clear, in my opinion.

I am ready to accept the manuscript, however I would only like the authors to take these final considerations into account, reported below. More specifically, for the manuscript to be as clearest as possible, I would like the authors not to refer to this phenomenon as “multiple stellar populations”, as it is known that the multiple populations phenomenon is the one associated with the old GCs (the chemistry variations). Find below all the changes and some suggestions for replacement:

AUTHORS:

We specified that we refer to extended Main-Sequence Turn-Offs and split Main-Sequences whenever possible. However, we believe that removing any references to multiple stellar populations would shift the scope of the paper, and, therefore, we opted to maintain “multiple populations”.

Abstract

Line 9 - remove “multiple stellar populations” -> The origin of star clusters , or young star clusters

Line 19 - remove “multiple populations” -> “stellar populations in young clusters”

AUTHORS:

We prefer to maintain the use of multiple populations, as removing it would change the scope of the paper

Introduction

Line 23 - remove “multiple stellar populations” -> the discovery of multiple sequences etc

AUTHORS:

We modified as suggested

Line 37 - It could be useful to explain with a few words what are these multiple populations: “...with multiple stellar populations, i.e. star-to-star light-element abundance variations (10,11)”.

AUTHORS:

We modified as suggested

Line 66-67 - remove multiple stellar populations & Line 68- remove multipopulation phenomenon

You can possibly change this sentence to: “NGC 1818 represents the perfect target to unveil the origin of the eMSTO as it is very young and shows the typical features found in young Magellanic Clouds clusters' CMDs”.

AUTHORS:

We modified the sentence

Line 70 remove “multiple populations “ ->“main properties of young clusters”

AUTHORS:

We opted to maintain “multiple populations”, as removing it would change the meaning of the paper.

Line 73 remove “multiple populations”

AUTHORS:

We opted to keep “multiple populations”. The sentence would lose meaning without “ multiple populations”

Line 103 - remove “multiple populations” - replace with eMSTO - split MSs... or the multiple sequences in the CMDs of young clusters.

AUTHORS:

We modified as suggested

Line 111 - remove “multiple populations” - ... has never been used to estimate the star formation history of young and massive Magellanic Cloud clusters stellar populations.

AUTHORS:

We opted to maintain “multiple populations”. Indeed this approach has been used to estimate the star formation history of young star-forming regions (see references in the text), but this is the first time that it is exploited in the context of eMSTO and split MSs in Magellanic Clouds clusters.

Line 165 - remove “multiple”

AUTHORS:

We removed the term “multiple”

Line 171 - remove “multipopulations” - For example, replace with “star formation history” or the origin of the eMSTO

AUTHORS:

We replaced “multiple populations” with the origin of eMSTO

Also, the text has some problems of visualisation. The equations are not visible anymore.

AUTHORS:

There are some visualisation problems when converting the file from a google doc one to word or similar. We tried to solve the issue.

Fourth Report
REVIEWERS' COMMENTS

Reviewer #2 (Remarks to the Author):

I really appreciated the new version of the paper which is much more solid and clear, in my opinion.

I am ready to accept the manuscript, however I would only like the authors to take these final considerations into account, reported below. More specifically, for the manuscript to be as clearest as possible, I would like the authors not to refer to this phenomenon as “multiple stellar populations”, as it is known that the multiple populations phenomenon is the one associated with the old GCs (the chemistry variations). Find below all the changes and some suggestions for replacement:

AUTHORS:

We specified that we refer to extended Main-Sequence Turn-Offs and split Main-Sequences whenever possible. However, we believe that removing any references to multiple stellar populations would shift the scope of the paper, and, therefore, we opted to maintain “multiple populations”.

REPLY Referee:

The main point I raised throughout all these reports is that the scope of the paper has to be shifted in order for the paper to be accepted. Hence, I would prefer that the words “multiple populations” are removed from all the sentences as mentioned in the previous report (below). In the community there is a huge confusion because of the nomenclature used. The phenomenon that the authors are studying in this paper cannot be correlated, based on the results of this paper, to the phenomenon of multiple populations. The use of this term (multiple populations) will create even more confusion, hence I will not accept the paper if the changes suggested at the last report are not going to be adopted.

Abstract

Line 9 - remove “multiple stellar populations” -> The origin of star clusters , or young star clusters

Line 19 - remove “multiple populations” -> “stellar populations in young clusters”

AUTHORS:

We prefer to maintain the use of multiple populations, as removing it would change the scope of the paper

Introduction

Line 23 - remove “multiple stellar populations” -> the discovery of multiple sequences etc

AUTHORS:

We modified as suggested

Line 37 - It could be useful to explain with a few words what are these multiple populations: “... with multiple stellar populations, i.e. star-to-star light-element abundance variations (10,11)”.

AUTHORS:

We modified as suggested

Line 66-67 - remove multiple stellar populations & Line 68- remove multipopulation phenomenon

You can possibly change this sentence to: “NGC 1818 represents the perfect target to unveil the origin of the eMSTO as it is very young and shows the typical features found in young Magellanic Clouds clusters' CMDs”.

AUTHORS:

We modified the sentence

Line 70 remove “multiple populations “ ->“main properties of young clusters”

AUTHORS:

We opted to maintain “multiple populations”, as removing it would change the meaning of the paper.

Line 73 remove “multiple populations”

AUTHORS:

We opted to keep “multiple populations”. The sentence would lose meaning without “ multiple populations”

Line 103 - remove “multiple populations” - replace with eMSTO - split MSs... or the multiple sequences in the CMDs of young clusters.

AUTHORS:

We modified as suggested

Line 111 - remove “multiple populations” - ... has never been used to estimate the star formation history of young and massive Magellanic Cloud clusters stellar populations.

AUTHORS:

We opted to maintain “multiple populations”. Indeed this approach has been used to estimate the star formation history of young star-forming regions (see references in the text), but this is the first time that it is exploited in the context of eMSTO and split MSs in Magellanic Clouds clusters.

Line 165 - remove “multiple”

AUTHORS:

We removed the term “multiple”

Line 171 - remove “multipopulations” - For example, replace with “star formation history” or the origin of the eMSTO

AUTHORS:

We replaced “multiple populations” with the origin of eMSTO

Also, the text has some problems of visualisation. The equations are not visible anymore.

AUTHORS:

There are some visualisation problems when converting the file from a google doc one to word or similar. We tried to solve the issue.

AUTHORS:

We thank the referee for his/her comments which improved the quality of the manuscript. We reply to each point below.

REVIEWERS' COMMENTS**Reviewer #2 (Remarks to the Author):**

I really appreciated the new version of the paper which is much more solid and clear, in my opinion.

I am ready to accept the manuscript, however I would only like the authors to take these final considerations into account, reported below. More specifically, for the manuscript to be as clearest as possible, I would like the authors not to refer to this phenomenon as “multiple stellar populations”, as it is known that the multiple populations phenomenon is the one associated with the old GCs (the chemistry variations). Find below all the changes and some suggestions for replacement:

PREVIOUS AUTHORS REPLY:

We modified the text specifying that we refer to extended Main-Sequence Turn-Offs and split Main-Sequences, whenever possible. However, we believe that removing any references to multiple stellar populations would shift the scope of the paper, and, therefore, we opted to maintain “multiple populations” in a few places along the text.

REPLY Referee:

The main point I raised throughout all these reports is that the scope of the paper has to be shifted in order for the paper to be accepted. Hence, I would prefer that the words “multiple populations” are removed from all the sentences as mentioned in the previous report (below). In the community there is a huge confusion because of the nomenclature used. The phenomenon that the authors are studying in this paper cannot be correlated, based on the results of this paper, to the phenomenon of multiple populations. The use of this term (multiple populations) will create even more confusion, hence I will not accept the paper if the changes suggested at the last report are not going to be adopted.

AUTHORS: In the following, we list all the changes that are highlighted in blue in the main text. Specifically, we removed any reference to Multiple Populations, either by removing it or by explicitly replacing it with eMSTO and split MSs.

Abstract

Line 9 - remove “multiple stellar populations” -> The origin of star clusters , or young star clusters

Line 19 - remove “multiple populations” -> “stellar populations in young clusters”

AUTHORS:

We removed the references to Multiple Populations as suggested

Line 70 remove “multiple populations “ ->“main properties of young clusters”

Line 73 remove “multiple populations”

AUTHORS:

We removed Multiple Populations as suggested.

Line 111 - remove “multiple populations” - ... has never been used to estimate the star formation history of young and massive Magellanic Cloud clusters stellar populations.

AUTHORS:

We changed the sentence to ‘but has never been used, so far, to investigate the star formation history of young Magellanic Clouds clusters with eMSTO and split MSs.’ The changed sentence has been highlighted in blue in the main text.